# Early warning signals have limited applicability to empirical lake data

Duncan A. O'Brien [1] ✉, Smita Deb[2], Gideon Gal [3], Stephen J. Thackeray [4], Partha S. Dutta [2], Shin-ichiro S. Matsuzaki [5], Linda May [6] & Christopher F. Clements [1]

Research aimed at identifying indicators of persistent abrupt shifts in ecological communities, a.k.a regime shifts, has led to the development of a suite of early warning signals (EWSs). As these often perform inaccurately when applied to real-world observational data, it remains unclear whether critical transitions are the dominant mechanism of regime shifts and, if so, which EWS methods can predict them. Here, using multi-trophic planktonic data on multiple lakes from around the world, we classify both lake dynamics and the reliability of classic and second generation EWSs methods to predict whole-ecosystem change. We find few instances of critical transitions, with different trophic levels often expressing different forms of abrupt change. The ability to predict this change is highly processing dependant, with most indicators not performing better than chance, multivariate EWSs being weakly superior to univariate, and a recent machine learning model performing poorly. Our results suggest that predictive ecology should start to move away from the concept of critical transitions, developing methods suitable for predicting resilience loss not limited to the strict bounds of bifurcation theory.

Natural ecosystems can display abrupt and non-linear shifts in state and function[1,2] which impairs societal activities dependent upon them. If these abrupt changes are sufficiently large to lead to persistent local ecosystem degradation[3] and negative socio-economic impact[4], then they are often classified as 'regime shifts'[5]. Regime shifts themselves are, however, but one observable system behaviour (Fig. 1) with varying mechanisms of change possible[1,6,7], all impactful on ecosystem functioning. The specific concern around regime shifts is their anticipated increase in global frequency in response to climatic changes[8], with the potential to cascade across systems[9]. Managing ecosystems prone to these shifts is therefore vital, but doing so is challenging using current analytical tools. Resultingly, effective detection tools for characterising oncoming regime shifts are desirable, particularly cost-effective approaches accessible to all economic statuses.

Critical transitions (a.k.a catastrophic transitions—Fig. 1A) are regularly suggested as the dominant process driving regime shifts via tipping points and positive feedback loops[10–12]. From this understanding, a toolbox of techniques have been developed aiming to characterise oncoming transitions[13]. These so-called Early Warning Signals (EWSs) are derived from bifurcation theory, which describes how a system can flip between two or more stable states across a critical/tipping point. EWSs attempt to detect this critical point by the phenomenon of Critical Slowing Down (CSD) or the increasing return time to equilibrium following perturbations as a critical transition is approached[10]. CSD increases as the system loses resilience and its stability weakens[1]. Unfortunately, the success of EWSs has been mixed despite widespread interest and research effort[6]. For example, EWSs are consistently successful in simulated systems[7,14], whereas limited examples exist in

[1]School of Biological Sciences, University of Bristol, Bristol BS8 1TQ, UK. [2]Department of Mathematics, Indian Institute of Technology Ropar, Rupnagar, Punjab 140001, India. [3]Kinneret Limnological Laboratory, Israel Oceanographic & Limnological Research, PO Box 447 Migdal, Israel. [4]Lake Ecosystems Group, UK Centre for Ecology & Hydrology, Bailrigg, Lancaster, UK. [5]Biodiversity Division, National Institute for Environmental Studies, 16-2 Onogawa, Tsukuba, Ibaraki 305-8506, Japan. [6]UK Centre for Ecology & Hydrology, Bush Estate, Penicuik, Midlothian EH26 OQB, UK. ✉e-mail: duncan.a.obrien@gmail.com

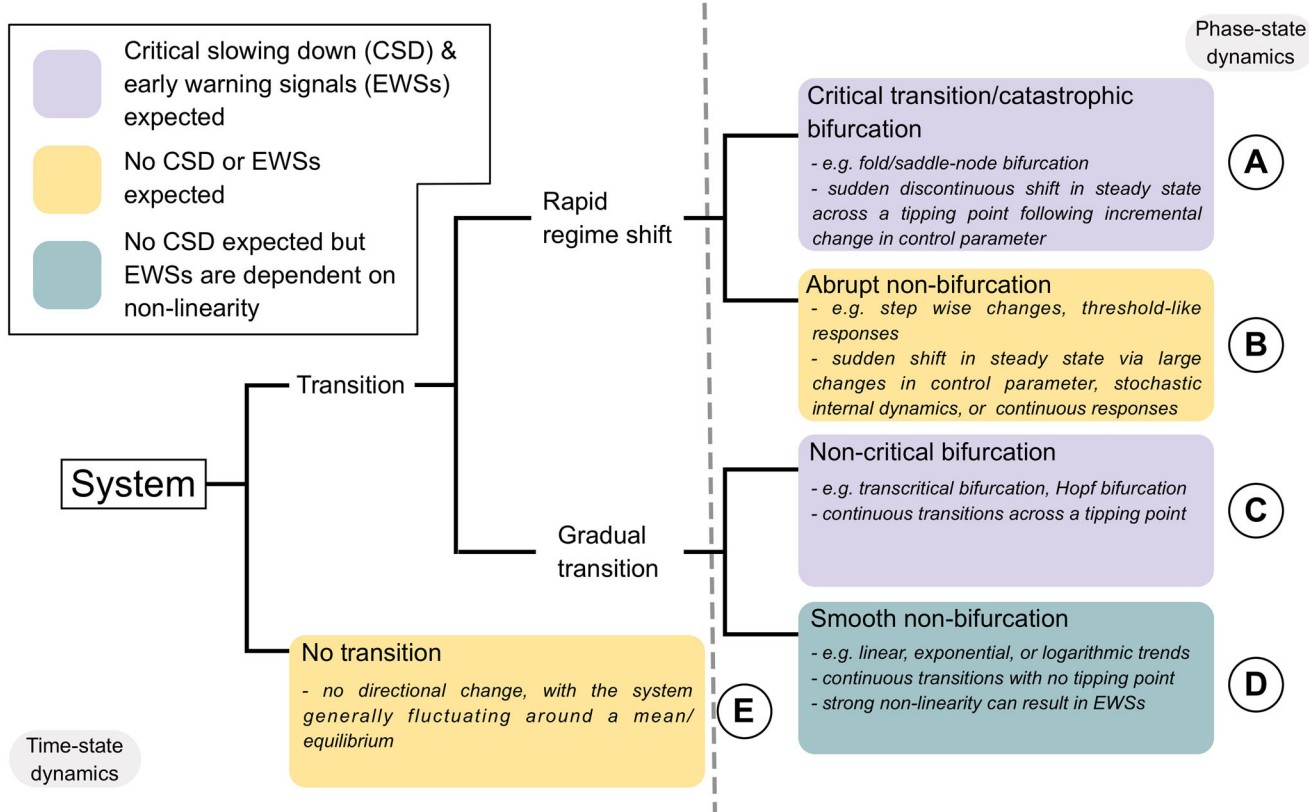

**Fig. 1 | Classification tree relating the primary transition types and system dynamics relevant to early warning signals (EWSs) and regime shift detection.** Terms are colour coded by their expectation to exhibit critical slowing down (the phenomenon quantified by EWSs). Whether a classification is made using the time series or the phase space (i.e. state against the control parameter/driver) is indicated. Distinct classifications are labelled (**A**–**E**) for crossreferencing with the main text.

natural settings[15–17]. Indeed, EWS assessments across empirical time series have been poor in their prediction of 'change'[18,19], leading to doubts about the practical utility of EWSs for ecosystem managers.

Major concerns of EWSs include their focus on 'transitioning' systems[20], data pre-processing[14], and, in our opinion, the assumption that critical transitions are common in natural systems[21]. Crucially, there is a difference in definition between 'regime shifts' and 'critical transitions' (Fig. 1, Table 1), but the two are often assumed the same[18,19,22]. This synonymy leads to confusion regarding whether most ecological regime shifts are critical transitions and, if not, are EWSs appropriate for generic regime shift assessment. In its simplest form, a regime shift is 'the process whereby an ecosystem rapidly changes from one fundamental state to another'[1,23] whereas critical transitions (Fig. 1A) are but one possible mechanism of regime shift which can only be inferred (though not confirmed without experimental or modelling approaches) from observational data if a number of criteria are satisfied[12,24–26]. These criteria include: (1) an abrupt shift in time series, (2) driven by positive feedback mechanisms, (3) in response to an incremental increase in control parameter, which (4) results in a multimodal distribution of state values (alternative stable states), and (5) displays hysteresis that limits the reversion of the shift. Alternative regime shift mechanisms are possible (Fig. 1B) and will display multiple criteria required for identifying a critical transition but will not include all of them. These mechanisms include stepwise changes in state resulting from a stepwise change in the control parameter (exhibits criteria 1 and 4), noise-induced transitions that occur despite no changes in control parameter (exhibits criteria 1, 2, and 4), or threshold-like transitions with sudden but continuous changes in response to gradual control parameter change (exhibits criteria 1, 2, 3, and 4)[1].

There is the additional concern that alternative stable states, and their tipping points of management interest, may be unsuitable concepts for ecological systems due to the influence of multiple stressors and system 'inertia' dampening/delaying regime shifts[27,28]. Neither lake chlorophyll[29] nor global metanalysis of stressor-system state relationships[28] suggest these systems consistently display multiple equilibria (criteria 4 for a critical transition) or tipping points (criteria 1 and 2), despite the current paradigm that lakes can exist in multiple stable states[30,31]. Many studies assessing EWS capability in empirical data[18,19] may, therefore, have been hindered as the indicators have been developed to solely quantify system dynamics under certain conditions. As such, more precise classifications of empirical transitions[24,25] are necessary to disentangle critical transitions from these alternative regime shift mechanisms[1] and allow robust assessments of EWS ability in empirical data.

A further complication for EWS users is that, theoretically, regime shifts do not occur uniformly across all taxa[32]. Most regime shift or alternative stable state research focus on single representative measures of the system such as total abundance[29,33], total cover[34] or biomass[13], when multiple measures are more insightful. Recent theoretical work in multi-species systems indicate certain life-history characteristics can result in taxa being more or less illustrative of CSD[32], while the interaction strength between taxa can mask transitions[35]. Reliance on univariate time series can therefore make it challenging to define an ecosystem's dynamics as stable or transitioning[36], particularly as most previous work neglects stationary time series[20]. Consequently, although the data is available, explicit assessment of real-world critical transitions across taxa, trophic levels and transition types independently is ultimately lacking.

## Table 1 | Glossary of terms

| Term | Definition | Reference |
|---|---|---|
| Regime shift | Sudden or abrupt shift in the state of the system resulting from the influence of an external control parameter/driver or by the system's internal dynamics, where core ecosystem functions, structures and processes are fundamentally changed. A regime shift may be associated with bifurcations (after crossing control parameter thresholds/tipping points), step changes in state (in response to step changes in control parameter), threshold-like responses (sigmoidal response to control parameter), or limit cycles (cyclic changes due to the system's internal dynamics). These abrupt shifts may also occur across different trophic levels. | 1,6,24,54 |
| Bifurcation | Gradual changes in a control parameter drive 'qualitative' change in the behaviour of an equilibrium point in a dynamical system. | 26,54 |
| Tipping point | A threshold value at which a dynamical system undergoes a sudden shift from one stable state to another alternative stable state in response to small stochastic perturbations. | 86 |
| Bifurcation point | A threshold value specifically associated with a bifurcation. | 6,86 |
| Critical transition/catastrophic bifurcation | A sudden shift from one steady state of a dynamical system to an alternate state via a fold/saddle-node bifurcation (a first-order or discontinuous transition). A discontinuous response follows incremental change in the control parameter crossing a critical value/bifurcation point. All critical transitions are regime shifts but the reverse is not necessarily true. | 6,7,86 |
| Non-critical transition/bifurcation | A non-catastrophic transition (a second-order or continuous transition) can occur via a transcritical, pitchfork or Hopf bifurcations across (a) bifurcation point(s). | 6,7 |
| Smooth transition | Continuous response to the control parameter in the absence of tipping points. | 7 |
| Critical slowing down | The phenomenon whereby the real part of the dominant eigenvalue of the system approaches to zero in the vicinity of a bifurcation point (and eventually goes to zero at the bifurcation point) while the return/recovery rate to equilibrium upon perturbation becomes increasingly slow. At the tipping or bifurcation point there is no chance of recovery. | 10,26 |
| Hysteresis | A system property the system may follow different paths when increasing or reducing a perturbation. Consequently, multiple stable states exist under the same control parameter value, and is more common in systems with fold/saddle-node bifurcations. | 54 |

Fortunately, major technical developments have been made in the field which aim to circumvent these limitations. These include new computation techniques[37], exploitation of information from multiple time series[38] and machine learning[39,40]. We, therefore, have access to three forms of EWS: univariate, multivariate and machine learning models. In brief, univariate EWSs assess population level transitions by quantifying the presence of CSD[13], multivariate EWSs pool information from multiple time series to yield a community/system level assessment of CSD[38,41], whereas machine learning exploits other unmeasured but learnt characteristics of time series to report the probability of transition[39,40]. Both univariate and multivariate machine learning models are possible, but to date, only univariate forms have been trained specifically for tipping point classification[39,40,42]. Multivariate EWSs should logically improve the reliability of transition prediction as the maximum information and data can be generically exploited with little-to-no required knowledge of the system. However, as with classical EWSs, these developments have generally only received testing in simulated systems[38,43] supplemented with cherry-picked empirical transition data[39]. There is, therefore, a knowledge gap on how modern techniques perform in data relevant to managers and against previous methods.

In this work, we classify regime shifts, critical transitions, non-critical transitions and stationary systems (Fig. 2) in nine long-term lake monitoring datasets (Fig. 3 and Table 2). We focus upon freshwater lakes as these systems are pivotal in the development of ecological theory and regime shift research[24,30] while also providing long term and high resolution data sufficient to disambiguate trophic levels and display regime shifts. From these lake classifications, we aim to identify the frequency of critical transitions across lakes and planktonic trophic levels (Fig. 4A) and appraise the practicality of generic EWS usage in both stationary and non-stationary time series. We also test EWS techniques' success rates (Fig. 4B, C) while optimising their potential strength via data pre-processing. The explicit classification of lake fate reveals that many accepted regime shifts are not critical transitions, with different trophic levels responding uniquely to environmental change. Most EWSs, therefore, perform poorly, although multivariate indicators are weakly superior to univariate.

## Results

### Lake data

Nine publicly available and long term lake datasets were accessed—Lake Kasumigaura[44,45], Lake Kinneret[46], Loch Leven[47,48], Lower and Upper Lake Zurich[49], Lake Mendota[50,51], Lake Monona[50,51], Lake Washington[52], and Windermere[53]—spanning a range of longitudes and environmental conditions (Fig. 3 and Table 2). Planktonic data and environmental variables were extracted, standardised (see Materials and Methods), and averaged to monthly and yearly resolutions before being separated into phytoplankton and zooplankton trophic levels. Following standardisation, plankton genus richness ranged from 3–57 phytoplankton genera (median = 22) and 2–9 zooplankton genera (median = 4) in each lake.

### Quantifying lake dynamics

Using threshold generalised additive models (TGAMs), we identified optimal break points in each lake's total phytoplankton and total zooplankton density through both time and the environmental 'state-space', and quantified bimodality in extension of the approaches of Scheffer and Carpenter[24], Andersen et al. [54], and Bestelemeyer et al. [25]. (see Materials and Methods). The state-space was represented by the first principal component of a principal component analysis of water surface temperature, nitrate concentration and total phosphorous concentration. When comparing estimated break points between the time series and environmental models with state bimodality, we identified coherence in Lake Kasumigaura's zooplankton, Lake Kinneret's phytoplankton, Lake Monona's zooplankton, and Lake Washington's phytoplankton (Table S1). As these matched our hypothesised behaviour in Fig. 2, we consequently classified these regime shifts as critical transitions (Fig. 4A). Many of the other lakes displayed breakpoints in their time series (e.g., Lake Mendota's zooplankton) but these were not matched in the environmental state space nor fulfilled the other requirements (e.g., no overlap of clusters to indicate hysteresis, Figs. 2, 4A and Supplementary Fig. S1) and were therefore classified as abrupt non-bifurcations if they also displayed bimodality. There are, therefore, two primary classifications relevant to regime shift detecting EWSs—critical transitions and not critical transitions.

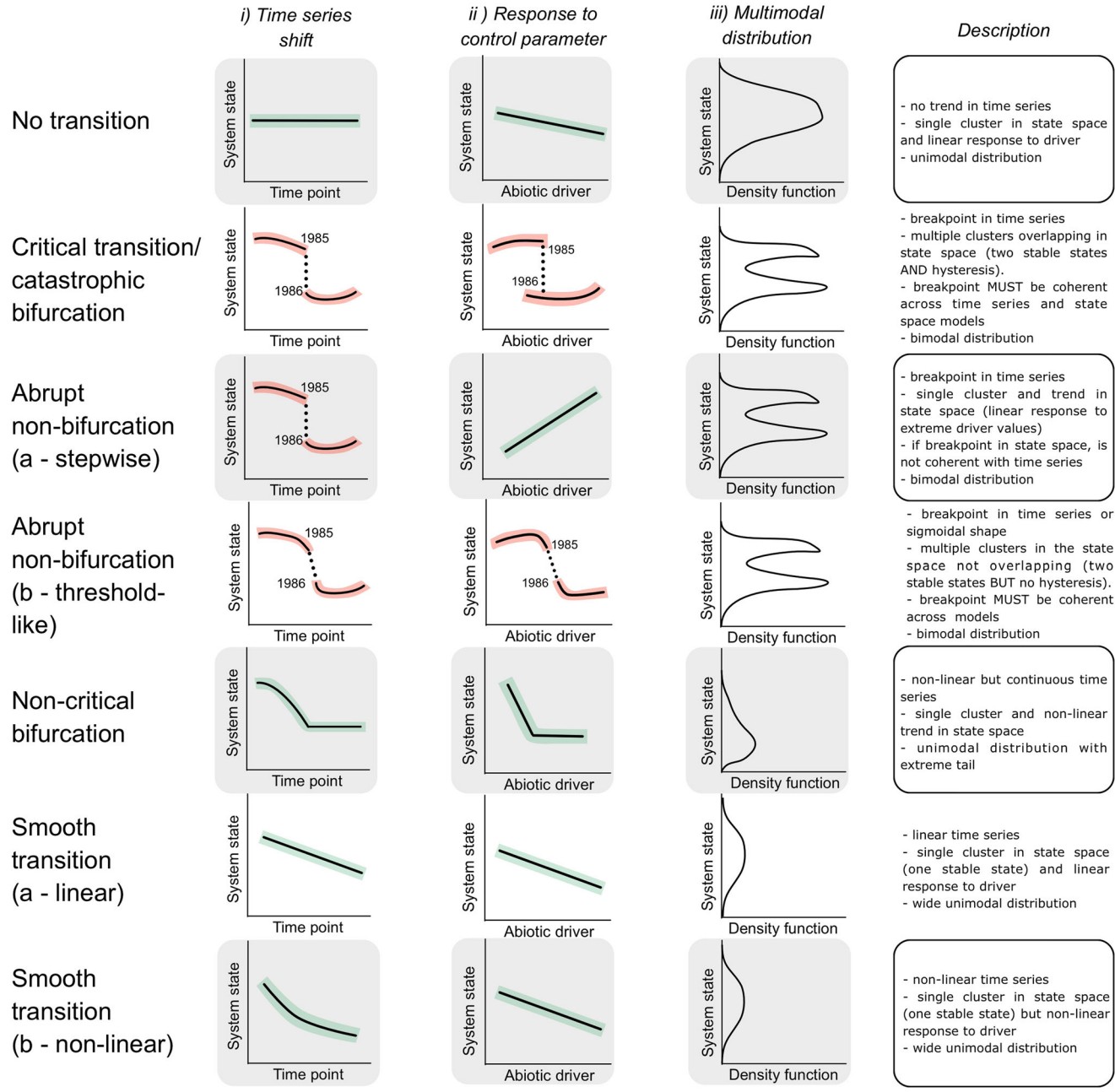

**Fig. 2 | Hypothesised behaviour of possible system dynamics under three complementary analyses used to classify the fate of a time series.** These analyses fulfil the criteria of Scheffer and Carpenter[24], Andersen et al. [54], and Bestelmeyer et al. [25]. for identifying alternative stable states in empirical data through (i) time series shifts, (ii) a hysteresis response to the control parameter and (iii) multimodal distributions. We have assumed here that the control parameter/ environmental driver is increasing through time. Analyses (i) and (ii) are performed using threshold generalised additive models (TGAMs) of plankton density against time and environmental driver respectively. Thresholds/breakpoints are only permitted to occur between adjacent time points. Analysis (iii) identifies unimodal vs bimodal distributions of plankton density across the entire time series. We expand these analyses over[24,25,54] to identify other forms of transition and provide a qualitative description for each transitions expected behaviour in the three analyses. In the first two columns, thick lines represent median TGAM fit with shaded regions the confidence interval. Dotted lines are discontinuities between breakpoints. In the third column, lines represent the density of observations. TGAMs are limited by classifying system dynamics solely upon observational data and, therefore will not guarantee classification without knowledge of the underlying system equations. Those equations can only be determined through experiments and differential equation modelling[24], but TGAMs provide a 'best-guess' using the limited data typically available to system managers.

---

Classifications made by TGAMs were then used to ground truth downstream EWS assessments and trim time series to pre-transition data. The time series classifications performed by TGAMs represent insights into the likely mechanism of change, covering a range of regime shift relevant mechanisms. There are, however, certain mechanisms that cannot be disambiguated without experimental or simulated work due to their similar behaviour across time and state-spaces. For example, threshold-like responses[54] and cusp bifurcations[7] will both display sigmoidal responses in both time and state-space (Fig. 2 Abrupt non-bifurcation b - threshold-like) but only the cusp bifurcation is anticipated to exhibit CSD. For this study, critical transitions are sufficiently different from other mechanisms to be classified (assuming some relaxation of driver has occurred to evidence hysteresis) compared to non-bifurcation regime shifts, but we suggest

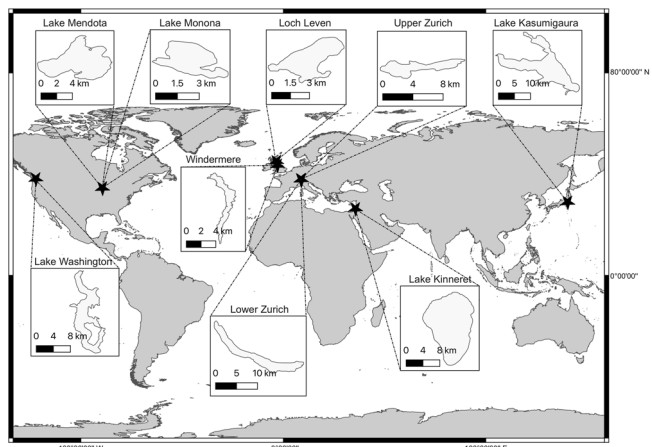

**Fig. 3 | Geographical locations of lakes assessed in this study and an indication of their size.** Map coordinates are projected in WGS 84.

that for qualitatively similar mechanisms, further evidence is necessary. Experimental and modelling to identify plausible system equations is an appropriate avenue to supplement TGAM fits to observational data only.

The final time series length for each lake consequently varied between 9–30 years (median = 17) depending on the TGAM identified transition year, grouped into 44 transitioning time series and 242 non-transitioning time series.

### Early warning signal pre-processing

Prior to EWS comparison, three detrending and deseasoning methods were performed to identify the optimal combination of detrending and deseasoning for each form of EWS. Specifically, we maximised the probability of correct prediction in the transitioning lakes, to represent the 'best case scenario' for EWSs. The detrending methods used included none, linear, LOESS (locally estimated scatterplot smoothing) and gaussian kernel, while deseasoning methods included none, average, time series decomposition and STL (seasonal trend estimation using LOESS). Each technique is regularly used in EWS assessments[13,55] and were applied to the data factorially. See Materials and Methods for more details.

We then used a Bayesian mixed effects binomial model to test the influence of each pre-processing technique on EWS ability relative to EWS assessments made upon the raw time series (i.e., none-none pre-processing). Details of the model structure can be found in the Materials and Methods. Across the five forms of EWS tested (univariate rolling window, univariate expanding window, multivariate rolling window, multivariate expanding window, and machine learning), each displayed optimal performance under different pre-processing methods (Table 3 and Supplementary Tables S3–S7). In univariate EWSs, rolling windows displayed the highest prediction probabilities under linear detrending and STL deseasoning (0.08 median estimate improvement over none-none detrending and deseasoning; Table S3) whereas expanding windows performed best under linear detrending and decomposition deseasoning (0.21; Table S4). Conversely, in multivariate rolling window EWSs, gaussian detrending and no deseasoning displayed the largest improvement over no pre-processing (0.21, Table S5), and gaussian-average pre-processing was optimal for multivariate expanding windows (0.24, Table S6). Finally, for the machine learning model, gaussian detrending and no deseasoning improved the probability of correct classification most relative to the raw time series (0.22; Table S7). Time series that underwent these pre-processing methods were then taken forward into cross-method comparisons (Supplementary Tables S8–S9), assuming the best-case data scenario.

### Early warning signal overall ability

Bayesian binomial models were then fitted to test the ability of each EWS computation method and individual indicators to correctly predict the transition fate (critically transitioning versus not) of the lake plankton time series. From this section onwards, binomial model estimates have been inverse-logit transformed from log odds to probabilities to improve interpretability. For raw model estimates, please refer to Supplementary Tables S8–S13 and Figs. S2–S10 for model diagnostics.

When assessments are pooled across indicators, lakes, trophic levels, and resolutions, multivariate EWSs estimated using expanding windows displayed the highest average probability of correct classification (Fig. 5 and Supplementary Tables S8 and S9). This probability is associated with these EWSs displaying the highest probabilities in monthly data (median [95% credible interval] = 0.59 [0.38–0.77], Fig. 5) and yearly data (0.71 [0.47–0.87], Fig. 5). Univariate EWSs estimated using expanding windows displayed the second highest probability in monthly data (0.57 [0.37–0.75]) and second highest in yearly data (0.67 [0.45–0.84). The remainder of computation methods were not strongly different from a 50% prediction probability although univariate rolling window EWSs were worse than chance in yearly data (0.29 [0.14–0.51]).

Prediction probabilities were more consistent across computation methods in monthly data than yearly, with a mean prediction probability of 0.51 ± 0.07 standard deviations compared to 0.49 ± 0.19. Univariate and multivariate rolling window EWSs especially declined in robustness when applied to yearly data relative to monthly. The machine learning model EWSNet was consistent across data resolutions at ~41%.

### Individual indicator ability

Splitting these computation method level trends into indicator trends within critical transition and not critical transition time series separately highlights how many of the above are driven by either strong true positive or true negative ability (Fig. 6 and Supplementary Tables S10–S13). For example, unscaled EWSNet displayed a 0.92 and 0.88 true positive probability in monthly and yearly data, respectively (Fig. 6A), but a 0.2 and 0.06 true negative probability (Fig. 6B). Scaled EWSNet displayed the inverse trend (true positive: monthly = 0.12, yearly = 0.08; true negative: monthly = 0.77, yearly = 0.97).

To clarify, we are only focussing on the median estimates here rather than the credible intervals due to the low replication of multivariate indicators. The model is weighted by the number of trials but for multivariate EWSs, time series are concatenated to a single assessment, inflating the uncertainty of multivariate indicators relative to univariate.

There was no coherent relationship between the method of EWS calculation (rolling versus expanding windows) across variates (univariate versus multivariate). Multivariate rolling window displayed the highest mean true positive probability when averaged across all indicators in monthly time series (mean ± standard deviation: 0.53 ± 0.20) and the highest probability in yearly time series (0.54 ± 0.16). Conversely, univariate expanding window EWSs were superior in not critically transitioning monthly time series (0.58 ± 0.10) as were multivariate expanding window EWSs in not critically transitioning yearly time series (0.63 ± 0.15).

This lack of coherence, therefore, suggests that individual indicators are highly variable and so should be considered individually. However, the dichotomy in prediction abilities observed for EWSNet was maintained across indicators and computation techniques. Composite univariate EWSs[56] computed via expanding windows (e.g., ar1 + SD, ar1 + SD + skew) were reliable across resolutions in not critically transitioning time series but maintained ~0.5 ability in critically transitioning time series. Autocorrelation at lag-1 (ar1) was the most reliable rolling window univariate EWS for critical transitions

**Table 2 | Data and monitoring characteristics of lakes contributing to early warning signal assessments**

| Lake | Sampling method | Sampling depth (m) | Original sampling frequency | Period | Monthly time series length post data processing | Yearly time series length post data processing | Number of genera | Regime shift identified in the literature |
|---|---|---|---|---|---|---|---|---|
| Lake Kasumigaura | Tube sampler and vertical net haul | 0–5 | Monthly | Aug 1981–Dec 2018 | 342 | 30 | 33 | ✓[57] |
| Lake Kinneret | Tube sampler, mix sampling and profile sampling | 0–40 | Weekly | Jan 1975–Dec 2015 | 289 | 25 | 31 | ✓[33] |
| Loch Leven | Tube sampler and vertical net haul | 0–5 | Weekly | Feb 1992–Dec 2006 | 152 | 12 | 6 | ✗ |
| Lake Mendota | Tube sampler and vertical net haul | 0–20 | Monthly | May 1995–Nov 2018 | 168 | 15 | 37 | ✓[58] |
| Lake Monona | Tube sampler and vertical net haul | 0–20 | Monthly | Apr 1999–Dec 2017 | 130 | 12 | 35 | ✗ |
| Lower Zurich | Tube sampler and vertical net haul | 0–135 | Monthly | Jan 1977–Dec 2009 | 332 | 28 | 40 | ✗ |
| Upper Zurich | Tube sampler and vertical net haul | 0–36 | Monthly | Jan 1980–Nov 2000 | 209 | 17 | 62 | ✗ |
| Lake Washington | Tube sampler and vertical net haul | 0–20 | Weekly | Jan 1962–Dec 1994 | 97 | 9 | 12 | ✓[52] |
| Windermere | Tube sampler and vertical net haul | 0–40 | Bi-weekly | Jan 1979–Dec 2002 | 244 | 20 | 16 | ✗ |

(Supplementary Tables S10 and S11). Multivariate rolling window indicators such as mean autocorrelation (meanAR), PCA variance (pcaSD) and the dominant eigenvalue of the maximum autocorrelation factor dimension reduction (eigenMAF) were particularly effective in transitioning time series but weak in non-transitioning data (Supplementary Tables S12 and S13). Overall, no indicator displayed reliability across resolutions or true positive/true negative time series better the 0.5 probability; eigenMAF displayed the highest cross scenario ability (monthly- true = 0.62; yearly-true = 0.70; monthly-false = 0.49; yearly-false = 0.52).

## Discussion

Motivated by debates surrounding multiple stable states in ecology and the need for reliable and generic critical transition detection tools, we assessed the prevalence of critical transitions in a range of empirical lake systems. We then compared the ability of current EWS methods to correctly predict ecosystem fate regardless of transition type or trophic level. We found that multiple regime shifts were identifiable across our lake network, but only a proportion of these were critical transitions. Similarly, within a lake experiencing a regime shift, phytoplankton and zooplankton trophic levels could display different regime shift mechanisms; one may critically transition whereas the other experiences a step change. Together, these findings may have hampered previous attempts to test EWS ability. Each EWS method was strongly influenced by the data pre-processing performed with no single indicator being reliable and no coherence across indicators. That said, multivariate EWS methods weakly outperformed univariate with the machine learning model EWSNet capable of extremely high true positive rates (though extremely poor true negative rates). Overall, computational and methodological advances are still not sufficiently reliable for generic usage due to their high sensitivity to data preprocessing and resolution, and conceptual concerns regarding the ubiquity of appropriate systems.

Generic EWS predictive ability is limited as it appears some system specific knowledge is necessary. Namely, understanding the potential for a critical transition/multiple stable states and identifying a mechanistic driver of transition are particularly key. These are not new arguments[1,20] but are worth reiterating to avoid the conflation of critical transitions with any form of abrupt change/regime shift. The TGAM approach used here, when combined with bimodality quantification, identifies the same regime shifts as previous studies[33,52,57,58] but only some of these were classifiable as critical transitions. We suggest it may be prudent to consider that temporal dynamics of driver variables (e.g., nutrient concentration) are themselves non-linear[59] and display pulse events[60] which can influence the mechanisms of regime shifts. For example, an anomalous year may push a system away from equilibrium into a long transient, as potentially occurred during 1985 in Windermere's zooplankton (Fig. 4), or a step change in environmental conditions can result in novel communities[1,61]. Consequently, critical transitions can occur earlier, later, or not at all, even if a regime shift occurs. This is compounded further as the disambiguation of critical transitions from certain other regime shift mechanisms can be complicated in empirical data[24,25,54], and the classifications we have made here are ultimately a 'best guess' given the data availability. For example, critical transitions (Fig. 1A) and threshold-like responses[54] (Fig. 1B) likely display identical time series, bimodality, and very similar state-space behaviour. The primary difference between the two mechanisms identifiable from empirical data is the presence of hysteresis which can only be observed if the system reverts entirely (i.e., regime shifts back to the original state) or partially (i.e., the driver relaxes back into the bistable region but not sufficiently for the system to shift back). As described in Fig. 2, hysteresis can be identified by an overlap of TGAM smooths in the state-space, while a threshold-like response has no overlap. All the lakes we classify as critical transitions do display some degree of overlap/hysteresis (Supplementary Fig. S1),

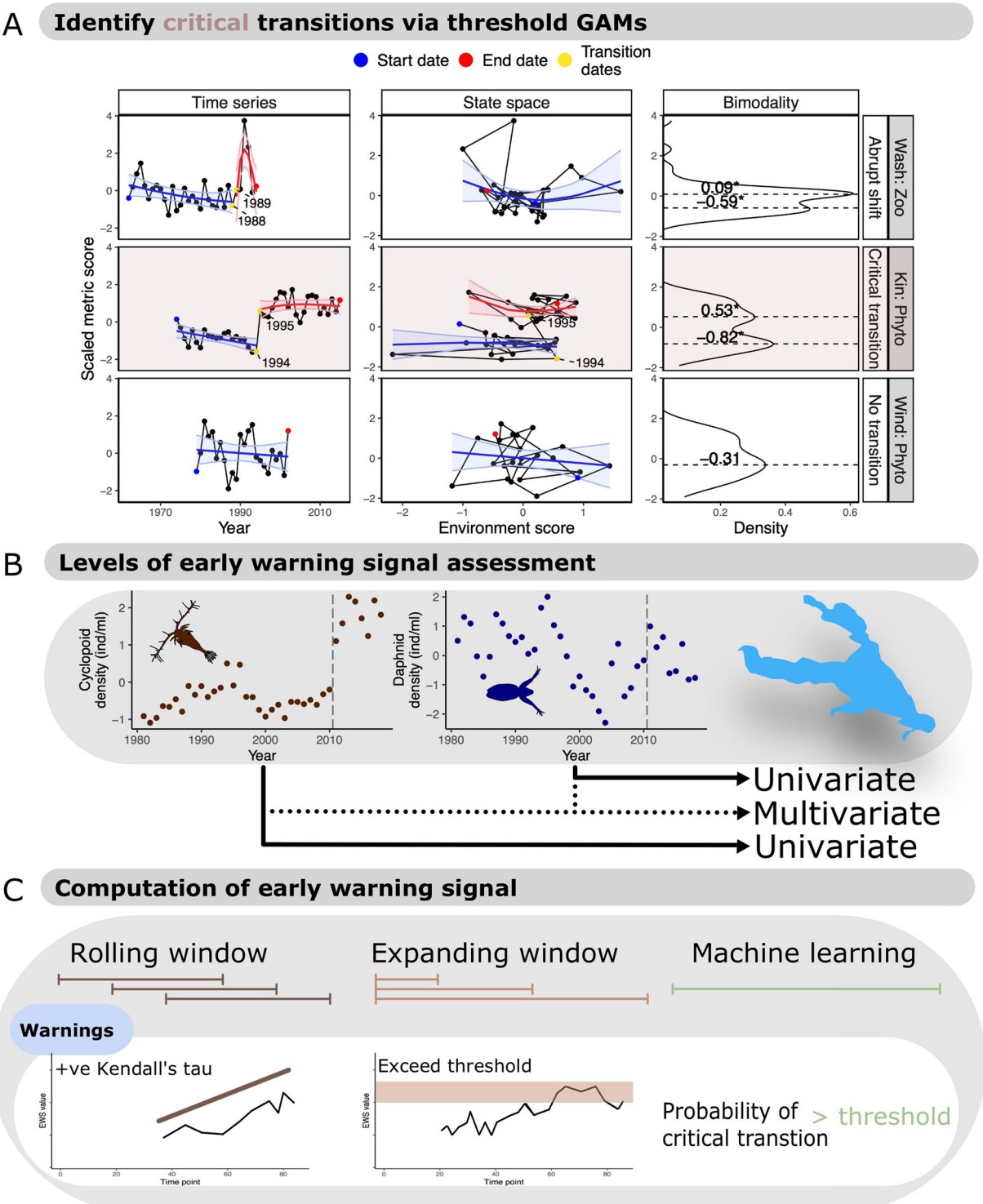

**Fig. 4 | Overview of generalised additive model (GAM) and early warning signal (EWS) techniques applied in this study, with selected examples from the lake plankton dataset. A** Application of the classification approaches introduced in Fig. 2 in the yearly lake plankton data. Points represent the observed data, with curved lines and shaded regions the GAM fits and 95% confidence intervals, respectively. Asterisks in the kernel density plot indicate significant bimodality coefficients, with dashed lines the estimated modalities. An example of an abrupt, non-bifurcation shift (Lake Washington's zooplankton), a critical transition (Lake Kinneret's phytoplankton) and a non-transition (Windermere's phytoplankton) are presented. Plankton densities have been scaled to mean zero and unit variance to improve plotting clarity. **B** Two forms of EWS are then performed on the plankton data: univariate EWSs, which consider one time series at a time, and multivariate, which combine information from multiple sources. The former can only give a representation of the community's state as assessment occurs at the species level, whereas the latter assesses at the community level. Here, yearly cyclopoid and daphnid densities from Lake Kasumigaura are presented. **C** The three computation techniques for calculating EWSs and 'warnings'. Rolling windows are the classical form of EWS and sequentially exploit a set proportion of the time series to calculate trends in the EWS—a strong positive correlation with time indicates an oncoming transition. Expanding windows incrementally introduce new data, with the rolling average of the EWS only signalling a warning if a threshold is transgressed. And machine learning models which predict the probability of a transition based upon its knowledge of its training data. The machine learning model used here (EWSNet) is limited to univariate time series whereas the other computation methods can be applied to univariate or multivariate EWSs.

**Table 3 | Optimal detrending and deseasoning combinations for each early warning signal computation method across time series resolutions**

| Data resolution | Early warning signal computation method | Optimal detrending method | Optimal deseasoning method |
|---|---|---|---|
| Monthly | univariate rolling window | linear | Seasonal and Trend decomposition using Loess (STL) |
| | univariate expanding window | linear | decomposition |
| | multivariate rolling window | gaussian | none |
| | multivariate expanding window | gaussian | averaging |
| | EWSNet (univariate machine learning model) | gaussian | none |
| Yearly | univariate rolling window | linear | NA |
| | univariate expanding window | linear | NA |
| | multivariate rolling window | gaussian | NA |
| | multivariate expanding window | gaussian | NA |
| | EWSNet (univariate machine learning model) | gaussian | NA |

but other lakes' breakpoints may not have sufficiently reverted for hysteresis to be identified. We therefore encourage empirical regime shift and EWS researchers to consider the mechanisms driving shifts[1,24,25] to maximise their reliability and appropriateness, and to not solely use EWSs as evidence of approaching tipping points.

These concerns are compounded by our finding that different components of the system (trophic levels here) can display distinct transition forms. Such incoherence is not unexpected, with certain taxa being more informative than others[32,35,62] but is rarely considered practically. Monitoring one representative variable is, therefore, unlikely to be sufficient and many facets of the system should be tracked, including plausible environmental drivers. In lake systems this is established practice but is more difficult to apply to other ecological biomes/scenarios. Drawing upon autonomous monitoring will contribute to this gap[63], but requires implementation.

Our EWS findings broadly agree with previous attempts to assess EWSs in aquatic systems[18,19] although those studies solely focus on classic univariate rolling window indicators. Indeed, our more explicit approach of disentangling critical transitions from regime shifts and intentional inclusion of stationary systems strengthens conclusions that classic EWSs struggle in empirical time series. We do, however, show that these EWSs can correctly classify non-transitions, a point not previously clarified due to an almost exclusive focus in transitioning data[20]. There are, however, strong methodological influences upon EWS ability, regardless of computation method, specifically through the choice of time series pre-processing, resolution of time series, and aggregation of time series to genus vs trophic level.

With real world time series, there is no ideal data for EWSs due to their typically high variability and cyclical nature driving false positive warnings[6]. Detrending and deseasoning are therefore necessary but are no silver bullet for accurate EWS assessments. Deseasoning is particularly complicated and capable of introducing spurious signals when the time series' value (i.e., plankton density/abundance here) are persistently close to zero. The sensitivity of EWS ability to data pre-processing reported here is therefore not unexpected[55,64] and arguably weakens EWS practicality. Choosing what 'system' EWSs are quantifying is also a key determinant of EWS ability. Here, we focus on the entire lake ecosystem rather than specific populations of interest (as occurs during fishery management[22,65]) but extrapolate that univariate signals from individual plankton genera are representative of trophic/lake level regime shifts. This is typical EWS usage[18,22,35] although we know different species/genera vary in their expression of critical slowing down[32,35]. Such taxon specific EWS behaviour limits the relevance of many genera in our lake dataset and influences our null findings from naïvely performing EWS assessments when no information is available to select specific taxa. Trophic and functional groupings appear valid levels to classify regime shifts, but EWSs require linear stability analysis (LSA) to identify representative taxa.

Unfortunately, many managers are unable to take this the approach due to the lack of calibrated lake/ecosystem models appropriate for LSA. We therefore urge caution when choosing where to apply EWSs. Multivariate EWSs mitigate some of the requirement for LSA but were not as successful as previously reported in simulated data[38,41].

The machine learning model EWSNet also performs variably depending on pre-processing–i.e., the scaling vs non-scaling of its training data. EWSNet's authors[39] advocate the finetuning of machine learning models with data of similar magnitude/dynamics as the target system to mitigate these issues. Here, we applied EWSNet generically without specific finetuning due to a desire to test its generic ability, plus challenges in generating appropriate training data for each lake system. This may explain the model's low classification success. That being said, if failure to detect a critical transition is more harmful than presumptuously intervening, then EWSNet in its generic form does have merit. We do overall, however, reaffirm previous suggestions that machine learning requires tailoring to the specific system of interest and is inappropriate for generic usage. Expansion to models trained upon multivariate bifurcation data may be an alternative solution. This begs the question however, whether if the system requires targeted modelling to finetune these signals, whether EWSs or machine learning can provide greater insight than such a model alone.

The combination of apparent EWS uncertainty and uncommonness of critical transitions strengthens the opinion that resilience/stability measures are a superior generic tool than CSD based indicators. While some of the measures tested here are considered stability indicators (e.g., mutual information[38]), more complicated measures have recently emerged independent from the assumption for local stability. For example, Ushio et al.[66] exploit empirical dynamic modelling to estimate the Jacobian matrix of a multivariate community and extract a stability index which accurately diagnoses vulnerable periods in fish communities. This has been developed further by Medeiros et al.[62] to identify key species for management based upon their contribution to the system's Jacobian and Grziwotz et al. for univariate time series[67]. Similarly, Williamson and Lenton's[68] approach Jacobian estimation using multivariate autoregressive models with equivalent success. Resilience/stability indicators have the additional benefit of not requiring post hoc detection of regime shifts nor critical transitions. Thus, practical real-time monitoring is possible regardless of critical transition risk as any unexpected loss of stability is a concern for ecosystem managers. EWSs can provide some information on resilience loss[1,38] but are conceptually linked to the presence of CSD[28]. Critical transitions should only conceptually occur when there are more than one state to transition across[1,28], and so EWSs should only be relevant in those circumstances. Stability indicators are ultimately capable of quantifying system stability both in proximity to critical transitions and in systems where transitions are not possible[69], lending

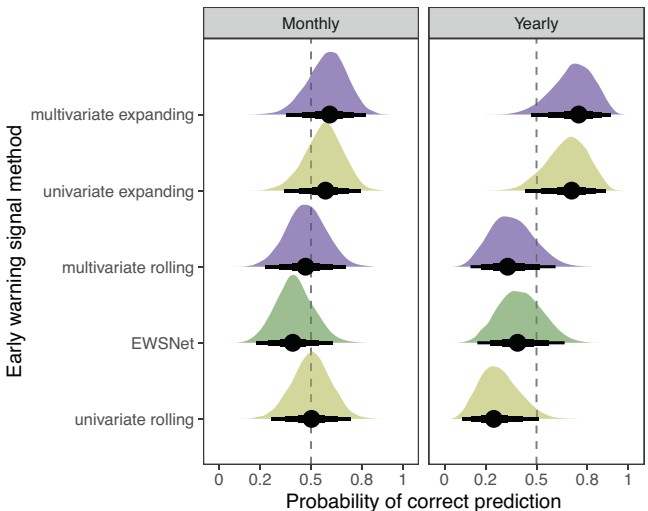

**Fig. 5 | Prediction success of lake plankton fate according to early warning signal computation method displayed as density plots of posterior distributions for time series level estimates of prediction ability grouped by computation method.** Computation methods are ranked by their mean ability across monthly and yearly data. The dashed vertical line shows the zero-slope−i.e., 50-50 chance of correct prediction−and each density plot represents 1000 samples from the posterior distribution of the parameter estimates. The reported values are the posterior density median values (circles), with 50% (thickest bars), 80% and 95% (thinnest bars) credible intervals back transformed from log odds to probabilities. Densities are therefore asymmetrical due to the sigmoidal relationship between log odds and probability.

them greater generic power than EWSs. Our findings support the conceptual use of these measures, due to the variation in EWS ability and diversity of lake dynamics observed.

To conclude, successfully applying and interpreting EWSs in real time is complicated and potentially unintuitive. Modern techniques struggle to be consistent and generic in empirical systems due each system's unique dynamics and lack of techniques for identifying true real-world critical transitions. Ecosystems can display regime shifts via many forms[1], many of which are theoretically not going to be detectable by EWSs. It therefore appears tailoring machine learning models to one's system of interest or focussing on resilience measures (which quantify stability rather than transitions) will be more useful for ecosystem managers. In systems where critical transitions are possible, combining information is critical to maximise EWS robustness but can still fail. The interpretation of EWSs generically applied across systems, therefore, requires caution.

## Methods
### Lake plankton data
Weekly/monthly plankton densities and abiotic variables were sourced from nine publicly available long-term monitoring datasets: Lake Kasumigaura[44,45], Lake Kinneret[70], Loch Leven[47,48], Lower and Upper Lake Zurich[49], Lake Mendota[50,51], Lake Monona[50,51], Lake Washington[52], and Windermere[53]. As these datasets encompassed a range of institutions, we performed a standardisation procedure. Unidentified and/or unnamed species were removed and if a species was not recorded on a sampling date, that species' density was assumed to be zero. This assumption results from constant search effort being made on each sampling date and unrecorded species being below detection threshold on that specific date. The data was then averaged to mean density per month and per year, allocated to trophic level (phytoplankton and zooplankton) and merged with three plausible abiotic drivers−water surface temperature (°C), nitrate concentration (µgL), and total phosphorous concentration (µgL).

## Critical transition pre-classification
For a sudden non-linear change in plankton density to be considered a critical transition, the system must display a sudden change in state following a small incremental change in the stressing variable mediated by positive feedback loops[10]. This should then manifest as multiple stable states (Table 1 and Fig. 2). Consequently, to identify historic critical transitions, assign system 'fates', and allow us to assess the ability of each EWS to classify transitions, we fitted threshold generalised additive models (TGAMs) to raw, yearly total (summed) plankton densities through both time and the environmental 'state-space', and quantified system bimodality (Figs. 2 and 4A) following Scheffer and Carpenter[24] Andersen et al.[54], and Bestelmeyer et al.[25]. Together, these three analyses allow us to disentangle critical transitions from other forms of non-bifurcation regime shifts such as pulse events or step changes, while also identifying non-linear but continuous transitions (i.e., non-critical transitions and smooth transitions), not feasible without the use of GAMs. That said, TGAMs are descriptive of observational data rather than diagnostic, and true classification requires some understanding of the governing system equations not achievable from observational data alone.

TGAMs fitted between time and plankton densities reveal sudden state changes (a.k.a. regime shifts), but the environmental model attempts to represent the environmental conditions the system is experiencing at each time point. Specifically, we represented these environmental conditions as the first principal component of the lake's abiotic drivers. We then used these TGAMs as a double validation tool where a breakpoint in the density time series indicates a non-linear step change in the system, whereas a breakpoint in the environmental state-space indicates a large change in response to a small change in stressor (Fig. 2). If there is also an overlap in state space smooths, this indicates hysteresis or a dual relationship is present for a single stressor value[24]. This was further verified using kernel density plots and bimodality coefficients[71] where a coefficient larger than 0.5 indicates a bimodal distribution. Therefore, we consider that a bimodal distribution and shared breakpoint between both the time series and state-space GAM fits is required for a critical transition to be identified (Fig. 2[24,25,54]).

We also performed our classification procedure independently across phytoplankton and zooplankton trophic levels as this dataset gives us the opportunity to question whether critical transitions are shared across both components of the system. For example, it is plausible that a critical transition in one trophic level will not necessarily be matched by a critical transition in the other if the former trophic level is a driver of the second; a critical transition in one system component drives a non-bifurcation regime shift in another. We did not perform classification at lower taxonomic or functional levels as, ultimately, we are interested in the prediction of regime shifts at the system/community level. For ecosystem managers, it is changes in functioning that is of concern[21,22] and it is the aggregate effect of system components that drives functioning[72]. Trophic levels represent the simplest linkage between functional groups and is the typical method of compartmentalising ecosystem models[30,73]. Classifying at the trophic level, therefore, is consistent with previous work and builds upon our general reconciliation of functioning with functional groups, or in their simplest form, trophic levels.

Practically, we followed the TGAM fitting procedure of Ciannelli et al.[74], where a breakpoint can be introduced into a GAM smooth. The optimal location of that break is identified by minimising the generalised cross-validation (GCV) score of the model (analogously to Akaike's information criterion), with the optimal choice between GAM and TGAM also selected via minimising GCV. To minimise the likelihood of overfitting, the total number of knots for each thin-plate spline smooth was restricted to a maximum of six, and, in the threshold form of the model, both halves of the smooth were

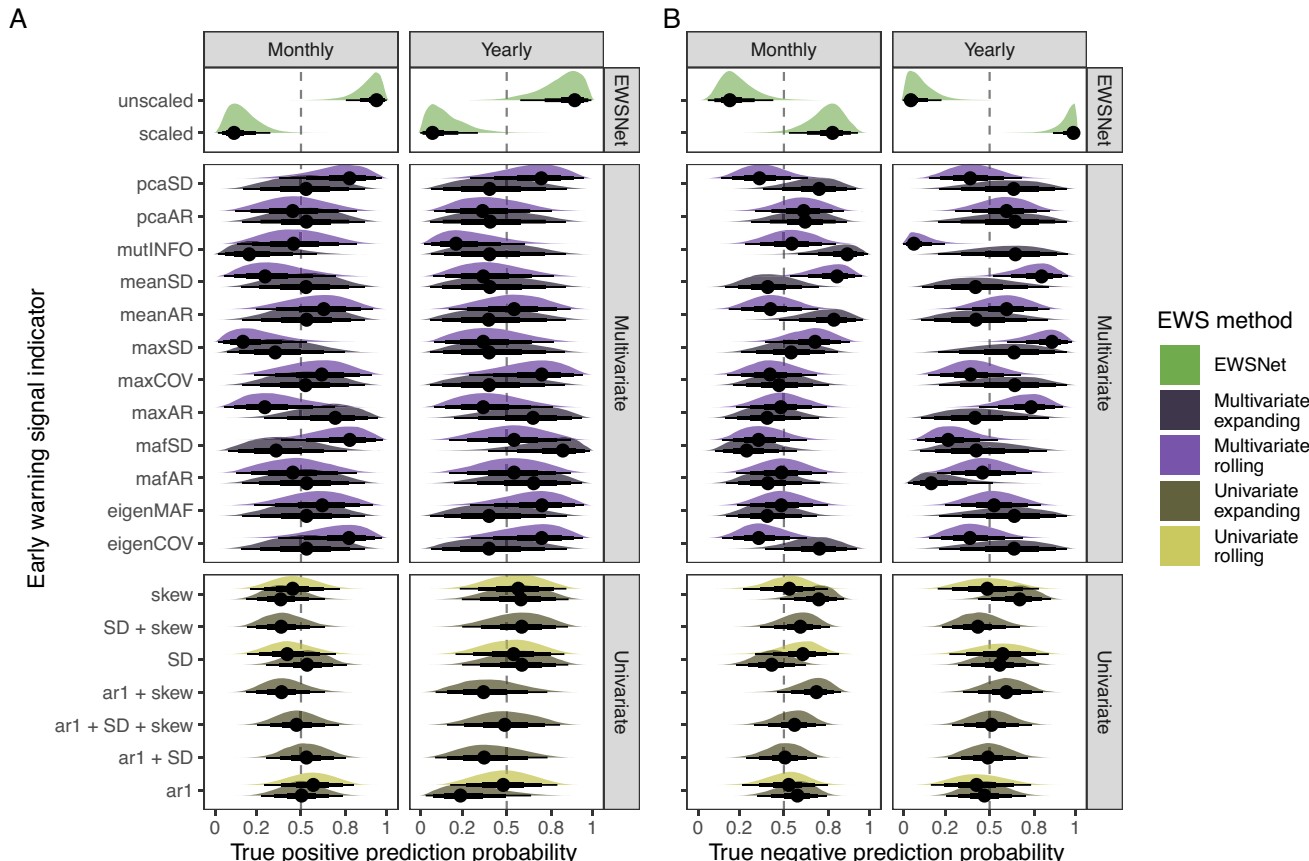

**Fig. 6 | Prediction success of lake plankton fate according to individual early warning signal indicators (abbreviations are defined in Table S2) displayed as density plots of posterior distributions for time series level estimates of prediction ability.** Estimates have been segregated into estimates made upon (**A**) transitioning data and (**B**) non-transitioning data. These models, therefore, represent indicator true positive prediction ability and true negative prediction ability, respectively. The dashed vertical line shows the zero-slope—i.e., 50–50 chance of correct prediction—and each density plot represents 1000 samples from the posterior distribution of the parameter estimates. The reported values are the posterior density median values (circles), with 50% (thickest bars), 80% and 95% (thinnest bars) credible intervals back transformed from log odds to probabilities. Densities are, therefore, asymmetrical due to the sigmoidal relationship between log odds and probability. Colour has been used to categorise the early warning signal computation technique (i.e., rolling window, expanding window, machine learning). Note—single estimates are reported for the univariate indicators ar1 + SD, ar1 + SD + skew, ar1 + skew, and SD + skew as these are only calculable via univariate expanding windows. All other indicators are calculable using both rolling and expanding windows.

restricted to three knots each. A threshold model was only fit if the continuous GAM smooth displayed an effective degree of freedom >= three. This represents an approximately cubic shape which plausibly contains a step change (as required for a regime shift) and minimises the likelihood of erroneously accepting TGAMs for approximately linear models. The optimal number of knots was selected via restricted maximum likelihood, and in both the time series and environmental models, breaks were only permitted between adjacent time points. All models were fitted through the package mgcv v1.8-40[75] implemented in R v4.2.1[76].

**Time series pre-processing**

Plankton species were then pooled to genus level to minimise the likelihood of zero densities, with a genus further dropped if they disappeared for a period longer than 12 months. This is necessary as downstream deseasoning can spuriously introduce cycles of non-zero densities into periods of zeroes. In addition, the strong phylogenetic signal in vertebrate and plant abundance trends implies that aggregating to genera level from species is unlikely to mask the strong CSD in critically transitioning taxa. The two final datasets of monthly and yearly temporal resolution for each lake consequently consisted of genus level densities across the two plankton trophic levels.

    Prior to early warning signal (EWS) assessment, each plankton time series was pre-processed via detrending with monthly time series

further deseasoned using a range of techniques. Detrending is considered important for improving the reliability of EWS assessments[55], so we applied three commonly used methods (linear detrending, LOESS smoothing, and Gaussian smoothing) and compared assessments made to those based upon the raw time series. Linear detrending fits a linear model between time and plankton density, with the residuals of this model representing the detrended time series[64]. LOESS, or local polynomial regression smoothing, subtracts a smooth curve fitted by local polynomial regression of span 0.5 from the raw time series[40], while gaussian kernel smoothing applies a linear filter, by subtracting the weighted moving average from the raw time series[72]. Additionally, monthly time series were deseasoned as monthly plankton data is inherently seasonal[77], and the repeated non-linear cycles can hinder EWS capability[78]. We, therefore, applied three deseasoning techniques (averaging, additive decomposition, and STL) factorially with the detrending methods to identify the optimal combination. Averaging simply subtracts the average value for a given month from the current data point of that month[79], additive decomposition estimates the seasonal cycle from moving averages which is then subtracted from the raw time series[80], and STL (seasonal trend estimation using loess), which also estimates the average seasonal cycle but uses local polynomials rather than linear/moving averages[81]. All data pre-processing was performed using the EWSmethods R package v1.2.0[82].

## Early warning signal assessments

Each plankton time series was then subjected to EWS assessments using each of the univariate, multivariate, and machine learning techniques provided in the *EWSmethods* package[82]. Supplementary Table S2 details the specifics of each indicator, but overall, univariate EWSs accept individual time series and quantifying the degree of critical slowing down (CSD) for that one taxon. Multivariate EWSs expand these assessments from single time series to multiple by either averaging across univariate EWSs or by extracting CSD information from a dimension reduction of the system (Fig. 4B). Here, we averaged performed dimension reductions across all time series within a trophic level to match the classifications made in the *Critical transition preclassification* section.

Finally, machine learning involves a convolutional neural network being trained upon mathematical models that undergo critical and non-critical transitions, to learn characteristics of a critical transition and report the probability of transition.

Univariate and multivariate EWSs were also assessed using two different computation approaches: rolling and expanding windows (Fig. 4C). Rolling windows portion the time series into a fixed window length, which then 'rolls' along the time series, incrementally calculating the indicators of Supplementary Table S2. The Kendall tau correlation of these indicators then represents the quantity of interest, where a 'strong' correlation coefficient is indicative of an oncoming transition. In this study, we maintained a rolling window size of 50% the total length of the time series following Dakos et al.[13]. The alternative expanding window computation incrementally introduces new data after a set burn in period. Each indicator is standardised by subtracting its running mean from its calculated value at time $t$ before division by its running standard deviation[37] using the equation:

$$EWS_t = \frac{ews_t - \overline{EWS}_{1:t}}{sd(EWS_{1:t})} \quad (1)$$

where $t$ is the current time point, *ews* is the estimated early warning signal indicator value across all data up to $t$, and *EWS* is the standardised running EWS value of all previous time points. A composite metric can then be constructed by summing all individual indicator values calculated per $t$. An oncoming transition is identified when the indicator/composite metric exceeded its expanding mean by a certain threshold value. Here, we set that threshold at two standard deviations due its favourable performance relative to alternative threshold values[65]. We also imposed a burn in period 50% the total length of the time series to mitigate spurious signals that occur at the beginning of assessment resulting from few data points in the window.

We also included the univariate machine learning model EWSNet[39]. EWSNet utilises the entirety of the pre-transition time series to provide probabilities of the likelihood of (1) a critical transition, (2) a smooth transition (please note this class in analogous to non-critical transitions defined in Table 1 and Fig. 1), or (3) no transition. We used the full ensemble of 25 models provided by EWSNet and averaged across them to improve the robustness of probability estimates following the suggestions of O'Brien et al.[82]. We additionally tested the effect of scaled versus unscaled data processing on the quality of EWSNet predictions. The scaled model involves training on the same data as the unscaled, but time series were normalised between within the range [1–2] using the following equation:

$$s = 1 + \frac{x - x_{min}}{x_{max} - x_{min}} \quad (2)$$

where $x$ is the training time series. This scaling, therefore, ensures all dynamics are considered at the same magnitude and aims to minimise the impact of measurement scale on predictions. When testing using the scaled form of EWSNet, the test time series must also be scaled for appropriate predictions.

To enable comparability between transitioning and non-transitioning taxa, lakes containing transitions were subset prior to the year identified by TGAMs. Resultantly, if only one of a lake's trophic levels experiences a critical transition, then all time series are subset prior to any regime shift. This minimises the likelihood of false positive signals driven by the changes in variance experienced in non-bifurcation regime shifts. Lakes with no regime shifts were subset to 85% of their total length. This ensures we can infer the near future of the non-transitioning lake correctly.

Additionally, as the various EWS method classes all generate different outputs, we converted these outputs into the binary presence-absence of a 'warning' (Fig. 4C). For rolling window computations, a warning was accepted if a positive Kendall tau correlation was in the 95th quartile of Kendall tau correlations from a dataset permuted from the original time series[13], for expanding windows when the two standard deviation threshold was exceeded for two or more time points[83], and for EWSNet, when the model predicted a critical transition (i.e., the strongest probability)[82]. This presence-absence of a warning was then compared to the ground-truth labels identified by the TGAMs, resulting in a binomial dataset of successes and failures. We only considered critical transitions here as these represent the primary classification of concern, due to their abrupt and hysteretic nature.

## Early warning signal ability

To estimate the classification ability of each EWS method, we developed a series of Bayesian hierarchical models using success/failure as response variable. Early warning signal method class and the specific EWS indicator itself were explored as categorical fixed effects in separate models: EWS method class (Supplementary Tables S8–S9) and indicator (Supplementary Tables S10–S13) ability. Early warning signal method class was treated as a factor with five levels: univariate rolling, univariate expanding, univariate machine learning, multivariate rolling, and multivariate expanding. Indicator was also treated as a factor with 21 levels for each indicator detailed in Supplementary Table S2. To account for the non-independence of repeated measurements for each lake and fate within each lake (transitioning versus non-transitioning trophic levels), we included a nested random effect of fate within lake identity. When testing the success of individual EWS indicators, we modelled ability across critically transitioning and non-critically transitioning time series separately to allow true positive and true negative indicator estimates. We did this to interpret whether overall EWS method class was influenced by good ability in transitioning data and poor in not transitioning, or vice versa, analogously to receiver operator curves (ROC)[6,40].

We also tested which combination of detrending and deseasoning methods maximised the probability of a correct prediction for each indicator in the transitioning lakes (Supplementary Tables S3–S8). This was represented as a factor with 16 levels for each pre-processing combination. We then used this information to fit the above models using time series which underwent the optimal combination.

The resulting general model structure was:

$$y_{ijk} \sim binomial\left(number\_of\_trials, \pi_{ijk}\right)$$
$$logit(\pi_{ijk}) = \beta_{factor} + u_j + u_k + u_{jk} + \varepsilon_{ijk} \quad (3)$$
$$u_{jk} \sim Normal(\alpha, \sigma_{jk})$$

where $y_{ijk}$ represents the log odds of correctly classifying a system's fate in the $i$th time series, in the $j$th lake with the known fate $k$. $\pi$ therefore represents the probability of the classification, $\beta$ represents slopes, $\mu$ the random intercepts, and $\varepsilon$ the remaining error. Each model was fitted without a global intercept to allow us to interpret the

absolute effect of each factor level on the probability of correctly classifying a time series' fate.

We set the weakly informative priors:

$$\beta_{factor} \sim Normal(0, 1.2)$$
$$-5.5 < \beta_{factor} < 5.5$$
$$\alpha \sim Normal(0, 1.2) \qquad (4)$$
$$\sigma_{jk} \sim exponential(1)$$

where *factor* represents the varying factors tested and *j* and *k* are lake identity and lake fate (the presence/absence of a critical transition identified by TGAM analysis), respectively. $\beta_{factor}$ is constrained between −5.5 and 5.5 as this represents a -1% to ~99% probability.

Models were written in the Stan language[84] and implemented in the brms v2.18.0[85] package, where they were run for 10,000 iterations following a 2000 iteration warmup period. Convergence was assessed via the identification of well mixed trace plots (Supplementary Figs. S2–S7) and appropriate Rhat values (equal to 1, Supplementary Tables S3–S13), with posterior predictive checks validating the final model posterior shape relative to the observed data (Supplementary Figs. S8–S10). During interpretation, we back-transformed the log odds into probabilities of correct classification, and used the overlap of the posterior distribution's credible intervals against 50% to identify EWS approaches that provide better estimates than chance. Modelling the probability of correct classification in this way allows us to control for confounding factors in the dataset, namely lake identity and the varying number of trials (i.e., EWS assessments) between lakes and EWS methods. This control is not possible using the F1-statistics and ROC typically used for binary classification tasks[39,40] which are limited to weighting based upon unequal sample sizes and cannot estimate co-dependencies between the repeated measurements inherent to ecological data.

**Reporting summary**
Further information on research design is available in the Nature Portfolio Reporting Summary linked to this article.

## Data availability
The raw plankton data used in this study are mostly available in the NERC Environmental Information Data Centre database under the identifier codes [https://doi.org/10.5285/de5ca6cc-02e9-42bc-a39e-80ec8acbffba; https://doi.org/10.5285/014f1c48-0838-49ca-b059-f084b13f4d5f; https://doi.org/10.5285/1de49dab-c36e-4700-8b15-93a639ae4d55], the Environmental Data Initiative database under the identifier codes [https://doi.org/10.6073/pasta/364622a6632f857289f9abc6a99d3ae7; https://doi.org/10.6073/pasta/6fc6015c620056034512fde089d50c27], and in the literature under the identifier codes [https://doi.org/10.1371/journal.pone.0110363; https://doi.org/10.3389/fmicb.2019.03155]. The raw plankton data for Lake Kinneret and Lake Kasumigaura are available under restricted access due to data ownership, access can be obtained by contacting GG and S-iM. The processed early warning signal data are available in the Zenodo repository [https://doi.org/10.5281/zenodo.10062493].

## Code availability
All code for analysis is available in the Zenodo record [https://doi.org/10.5281/zenodo.10062493].

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

## Acknowledgements

D.A.O. received funding from the GW4 + FRESH Centre for Doctoral Training in Freshwater Biosciences and Sustainability (NE/R011524/1), S.D. received funding from the Ministry of Education, Government of India (Prime Minister's Research Fellowship) and CFC received funding Natural Environment Research Council (NE/T003502/1 and NE/T006579/1NE/R016429/1). We thank Tamar Zohary for the long-term phytoplankton dataset from Lake Kinneret, Francesco Pomati for data from the Zurich lakes, the North Temperate Lakes LTER group for data from North American lakes, and Heidrun Feuchtmayr for Windermere data; monitoring at Windermere is currently supported by Natural Environment Research Council award number NE/R016429/1 as part of the UK-SCAPE programme delivering National Capability. The Lake Kinneret monitoring programme is funded by the Israeli Water Authority. We also thank the field and laboratory teams who have collected all of the data used in this study. This research was supported in part by the International Centre for Theoretical Sciences (ICTS) for the programme "Tipping Points in Complex Systems" (code: ICTS/tipc2022/9).

## Author contributions

D.A.O. performed the conceptualisation, methodology, investigation, software writing, formal analysis, writing-original draft preparation, and visualisation. S.D. supported the methodology, provided software validation, and review and editing of the manuscript. G.G. and S.J.T. contributed long-term monitoring data, review and editing of the manuscript and supervision of D.A.O.; P.S.D. contributed to the methodology and review and editing of the manuscript, and supervised S.D.; S.-i.S.M. and L.M. contributed long-term monitoring data and review and editing of the manuscript. C.F.C. was involved in the methodology, review and editing of the manuscript, and supervised D.A.O.

## Competing interests

The authors declare no competing interests.
