## [Peer Review File · Nature Communications]

REVIEWER COMMENTS

Reviewer #1 (Remarks to the Author):

General comments

In the article "Early warning signals are hampered by a lack of critical transitions in empirical lake data", the authors use time series from 9 lakes and different trophic levels (zooplankton, phytoplankton) to test and compare the ability of various early-warning indicators to predict critical transitions. The tested EWSs belong to three groups: classical univariate indicators, multivariate indicators and machine learning-based classification. In addition, the performance of the indicators was evaluated in combination with different preprocessing approaches for detrending and deseasoning and different computational approaches (rolling window, expanding window).

I find the manuscript is of general interest since the value and applicability of EWSs are still under debate. The contribution of this manuscript to the debate lays particular in the inclusion of different trophic levels, true positive and true negative transitions, different groups of EWSs, and the systematic investigation of the influence of different preprocess and computational approaches on the performance of the EWSs.

However, in the present form, I also have several points of criticism regarding the manuscript.

Most importantly, this regards the definition and used approach to identify critical transitions, which uses only one necessary but not sufficient criterium. Thus the critical transitions are not uniquely identified here, and the proclaimed ground truth in itself thus is highly uncertain. It is difficult for most rapid regime shifts found in empirical time series to rigidly differentiate between "State thresholds" and "Driver-state hysteresis" (following the terminology used by Andersen et al., 2009, Ecological thresholds and regime shifts: approaches to identification) and, in fact, is impossible without the presence of several shifts (see, for example, Scheffer et al. 2003 Catastrophic regime shifts in ecosystems: linking theory to observation).

Further, the author's claim to identify the ubiquity of critical transition needs to be toned down considerably since, firstly, as stated above, the authors can not claim to have identified critical transitions uniquely. Secondly, in the face of an estimated 304 million lakes (sensus Downing et al. 2006), 9 lakes, not even covering all climate zones, lake types etc., are nowhere near showing the ubiquity or none ubiquity of critical transitions.

Finally, the manuscript is not always easy to follow and evaluate. This is mainly due to a fuzzy use of core terminology, gaps in the method part and the distribution of information on the used method between the result and method section.

Therefore the authors should take care of a stringent definition early in the text of the core terminology around (rapid) regime shifts, bifurcation and critical slowing down. Preferably this definition is based on an accepted use in the community. I suggest, for instance, Boettinger et al., 2013, Early warning signals: the charted and uncharted territories. After such a definition, it is helpful for the reader and avoids misunderstandings to use this terminology consistently throughout the text. Here it could be helpful to add an accompanying diagram showing how terms relate, e.g. in the style of Boettinger et al. Fig. 1. I acknowledge some attempts to provide definitions by the authors (e.g. L49 ff, L74 ff); however, the definitions are not stringent in themselves and are not consistently used throughout the text. To exemplify the above point, I list some of the plethora of terms appearing in this text: abrupt non-linear changes, regime shifts, critical transitions, type of transition, whole ecosystem change, different forms of abrupt change, bifurcation-theory, process-driven regime shifts, tipping points, critical point, positive feedback loops, critical slowing down, alternative stable state, pulse events, step changes, smooth transition, step change/transient.

Regarding the deficits in the methods section, I would like a more detailed and differentiated description of the pre-processing steps carried out regarding the monthly and annual time series. In addition, it remains unclear how the breakpoints identified in the annual resolution are applied to the monthly time series. In this context, I am also missing a critical discussion of why regime shifts have been identified at the zooplankton and phytoplankton level rather than the genera level and how this may have affected the ability to detect regime shifts, particularly because regime shifts do not necessarily affect all species/genera with the same quality, as the authors themselves state.

Moreover, although testing the expanding windows approach is an important aspect of the manuscript, details on the method remain unclear, even after following the given citations.

Most importantly, details are missing on how the authors derive the central success/failure data table, which would be necessary for reproducibility. In particular, it is unclear how results from univariate EWS indicators, with one result per genus level, were made comparable to results from multivariate EWS indicators, which return one result for multiple time series (system).

Finally, the discussion is extensive and should be shortened to make room for critical assessments of assumptions made in the methods part (see, for instance, the points raised above).

See the specific comments section for minor comments and examples of the main criticisms above.

Specific comments

Abstract

L27: "... often inaccurate..." This is a strong claim since most EWSs have a solid theoretical grounding. So the question would be: inaccurate in which regard?

L31: From theory, it is clear that EWSs derived from CSD are not expected to predict all types of (rapid) regime shifts. Already here, I found the fuzzy use of terminology confusing.

L32,L35: "... identify the type of transitioning ..", "... different forms of abrupt change...". Be more precise. What transitions (group of transitions) do you distinguish (critical transition vs no critical transition, Table S1)

L37-38: Given that which group of EWS indicators performs best seems highly dependent on the temporal resolution of the time series and the chosen preprocessing and calculation approach, I don't see how you arrive at this general summary/conclusion.

L40: "... predict change ..." This is very generic - better "abrupt regime shifts" (see general comments regarding terminology).

Introduction

L76-78: No, this is a necessary but not sufficient criterium.

L84: Be more precise. Which ones?

L89: "uses" instead of "users"

L94: I had trouble finding this statement in the cited article; however, if this is the case, this also leads to uncertainty in your transition type classification, which needs to be critically discussed.

L123-125: Given that you are investigating only 9 lakes and your applied classification method can not uniquely determine a critical transition, it is necessary to tone down this statement.

L 125-126 Given the versatility of your results regarding preprocessing, temporal resolution, true positives, and true negatives, I would strongly recommend refraining from such generic, off-hand summaries.

Methods

L382: How do downstream deseasoning and detrending apply to yearly data?

L383-384: The disappearance of a species can be seen as an important change in community composition and can possibly also depict an abrupt change. How does the technical need to exclude those time series bias your data against (sudden) changes? Please critically discuss how some of your data processing steps could have affected your conclusions.

L385-386: Add: "The two final datasets on a monthly and yearly temporal resolution for each lake"

L390-391: How does this statement relate to yearly data?

L410-412: This is a necessary but not sufficient criterium!

L414, L154: If you identify a transition on yearly resolution, how did you divide the monthly time series into pre-and post-transitioning phases?

L416: How were plankton densities derived from the genius-level time series? As sums? How could this affect your ability to detect an abrupt shift (critical or otherwise), e.g. due to portfolio effects and not all genera might be affected equally by a regime shift? Also, here a critical discussion would be required.

L450-451: Over which group of time series where this averaged, zooplankton vs phytoplankton, i.e. trophic level? What is, in your case, the "system"?

L463-464: The calculation stays unclear, despite also following the cited articles since it is unclear how, e.g. autocorrelation at time t is calculated.

L473: Define "smooth transition". How did you map this to your "ground truth" (Table S1) for the performance evaluation?

L494: To follow this section, I would find it helpful if you would refer to the corresponding tables in the supplementary material.

L484-485, L491-L492: How exactly was it handled that for the univariate measures, there is one success/failure result for every genius level time series, while for multivariate methods, which are calculated across multiple time series representing "the system" (Table S2), there is only one.

L507: 15 levels.

L516-518: I can not fully reconcile this information to the one given in L171-172, particularly because also in tables S3-S7 there seems to be no factor level “none-none”, even so in the table headers “assessments made on raw data” is mentioned.

Supplementary material

A table of content would be helpful to navigate the material and get an overview of the various tests.

What is the temporal resolution of the time series for the results in tables S3-S7.

Results

L130 A map depicting the location of the lake would be appropriate. In addition, a table summarizing under which temporal resolution and type of grouping (overall, true positive, true negative) which group of EWS + preprocessing performed best would be helpful for the reader to keep track of the specific results.

L145-147, L150-151: As mentioned before, this is a necessary but not a sufficient criterium.

L185: Add reference to table S8, i.e. “ cross-method comparisons (table S8) ...”.

L184-185: How does this apply to the yearly resolution time series?

L192: “..tables S8-S13”.

L208: According to Figure 3, it should be “decreased”.

L225-231: Refer to the appropriate tables in the supplementary in this section.

L233: Specify “this”.

L234: Specify “the method”.

Discussion

The discussion in the present state should be shortened, and critical discussion to potential influences of methodological decisions added, particularly on the used “ground truth”, which was established on necessary but not sufficient criterium only.

L256-257: I do not see how you come to this clear conclusion, seeing the very mixed results. The performance and appropriate pre-processing seems situation dependent, e.g. temporal resolution of data; maybe a more differentiated recommendation would add value.

L340-342: Citation with evidence in regard to this.

Reviewer #2 (Remarks to the Author):

The paper by O’Brien et al “Early warning signals are hampered by a lack of critical transitions in empirical lake data”, as stated by the title and abstract, discusses observed variables from nine lakes, and compares multiple early warning signals.

The paper obviously contains a lot of results, but when the authors claim that this or that indicator performs better or worse in terms of detecting transitions, this is based on statistical characteristics rather than on ground truth information about transitions – this can be caused by parameters of fitted models and pre-trained setups.

Furthermore, many of the indicators may contain issues by construction: for example, multivariate indicators aggregate variables in a simple way, and some of those may have opposite dynamics, while others may be cross-correlated and lead to a bias in estimates such Kendall’s tau. This can affect the resulting multivariate indicators to such extent that any conclusion about its results may be misleading.

I wonder if the authors are familiar with works of Killick and their R package “changeoint”? <https://cran.r-project.org/web/packages/changeoint/index.html> If the authors aim to distinguish abrupt transitions, this is a suitable tool for such analysis. Furthermore, I cannot see references to true multivariate analysis (i.e., not just aggregates or averages) developed by Williamson and Lenton (Chaos, 2015). For comprehensive comparison, the authors should include these methods, in my opinion.

When the authors say that most published papers contain detected ecological transitions, this is an obvious effect of publication process: publishable are the results that contain a phenomenon of interest rather than those that do not. Publishing an absence of something is more difficult and is of marginal interest. I am sure the researchers in this community shelved lots of results where they detected nothing. I agree that such results can also be useful, but not necessarily for a broad research readership. In this context, in my opinion, the current manuscript is more suitable for a specialized ecological journal.

I always start reading a new paper by looking at its figures to see the data under study, and then at the obtained results, to see if they are non-trivial. With this paper, I started searching for the main data and, to my surprise, found it in the supplement (figure S1). I think this figure should be in the main text. Then, I looked carefully at the top panels of figure S1 and placed the ticks in the panels where I could detect some transitions by eye. These were panels 1, 2, 6 and 8 (lakes Kasumiguara, Kinneret, Monona, and Washington). Curiously, after reading the paper in full, I found that the authors' analyses detected critical transitions in the same datasets (Table S1). On the one hand, it is a good confirmation, on the other hand, it means that some less visible transitions may still be non-detectable by the applied methods. Therefore, modelled data is still of importance in studying such systems: it provides sufficient statistics and, more importantly, ensures controlled experiments with ground-truth comparison. I think the authors should change the title of their paper, as in its current form it sounds dismissive to the above issues. The title could be "Analysis of EWS in empirical lake data".

The paper from the start discusses ideas that are clearly defined only later (critical transitions and abrupt changes in lines 405-410, 415-420). I think such blocks of text should appear much earlier, in case if readers are not familiar with the topic.

When the authors distinguish "univariate, multivariate and machine learning" indicators, this may be misleading for people in the ML community, because they are aware of ML methods that can be both univariate and multivariate. It looks like the authors developed their own language in this context, but it is not widely accepted and is better to be avoided in a peer-review publication.

When the authors claim that they disentangle critical transitions from regime shifts (lines 292-295), this is well-known in the EWS community for years and usually is addressed by using different indicators/techniques. This is not a novel conclusion.

It would be useful to include a table with full description of the variables, sampling rates, durations, etc. Such a summary is important for overview of data. Also, include information about number of points in the original and processed data (monthly, yearly).

I do not agree that non-recorded points should be replaced by zeros (lines 375-376). This processing changes auto-correlations in the data, which is the basis of early warning signal indicators. Such pre-processing will change the indicator slopes.

It would be good to see a multi-panel figure of monthly data. In fact, detection of transitions in monthly data is more challenging and interesting, because noise would mask obvious transitions visible in yearly data – can the authors include such results? The provided panels with state variables are of less interest, in my opinion.

In lines 113-115, the authors mention nine lake datasets and refer to figure 1, which, in fact, does not contain real data (there are schematics). The sentence should be modified.

There are some terms and abbreviations that are not explained, like mgcv, brmc. I understand it is technical, but in that case, it can be in the supplement. There are acronyms that are defined multiple times (CSD, for example). It may be worth including a nomenclature.

In the Supplement, there are many plots with time series of chains (four overlapping records that are difficult to distinguish). I am not sure how useful is this, but it can remain in the supplement for information. The captions of figures S2-S7 mention monthly and yearly data, but it is not clear what lake is where – is data from all lakes combined? The y-axis labels are merged and unreadable. I think the label tick/label step should be changed when mapping the labels (there should be much fewer of them). In the right-hand-side panels, top labels are wider than the panels.

In tables S3-S13, why are effective sample sizes are not integer? Why in these panels R_{hat} is included, which is equal to 1 everywhere, - isn't it better to mention this in the text or caption instead of dragging the same value in a separate column in all tables?

The main text does not have page numbers, and the supplement has three pages numbered as 1.

While I appreciate the effort of processing data and applying multiple indicators in this manuscript, I think it requires further work and is more suitable for an ecological magazine.

Reviewer #3 (Remarks to the Author):

Summary of work

The authors conduct an analysis of early warning signals in empirical lake data. They obtain plankton densities and abiotic drivers from 9 lakes that have publicly available data, and assess whether a critical

transition occurs by fitting TGAMs and looking for break points in both the plankton and abiotic driver data. From a total of 244 time series, they find that 35 undergo a critical transition, by their definition. They test three different groups of early warning signals (EWS) on this data: univariate EWS; multivariate EWS; and EWSNet, a machine learning method. They also test a variety of different preprocessing steps (detrending and deseasoning), data resolutions (monthly and yearly averages), and EWS computation methods (rolling vs expanding window). They find that univariate EWS with an expanding window obtain the highest average performance in yearly data, and multivariate EWS with an expanding window obtain the highest average performance on monthly data. They found that other combinations of EWS were not much better than chance. They conclude that EWS based on bifurcation theory are not particularly useful in lake data, and that focus should turn to resilience indicators and methods that are specific to lake ecosystems.

Summary of evaluation

I applaud the others on conducting a thorough evaluation of a multitude of different EWS and preprocessing methods. It has already been demonstrated that EWS are not consistent in freshwater ecosystems (e.g. Gsell et al. PNAS, 2016), but this study builds on this work by proposing a new way to identify critical transitions in lake data, increasing the number of lakes analysed, testing EWS on transitioning and non-transitioning time series, and testing a wider range of EWS, including a recently developed machine learning method (EWSnet). I've since taken a look at the data myself, and I don't think it's surprising that the EWS do not perform well, based on the number of missing data points, the seasonality, and the relatively small number of data points prior to the transitions, but nonetheless, I think it's a useful contribution to the field, and an important word of caution to the use of generic EWS in empirical data.

I was able to download the code and data files from the Github repository, which is well documented. The code looks well organised, although I have not attempted to reproduce the results. The methods seem sufficiently detailed to follow the same steps as the authors. I think that the work is suitable for Nature Communications, although I have a few main comments and a few minor comments that I think should be addressed.

Main comments

- The processed dataset includes 35 'transitioning' and 209 'non-transitioning' time series. It's important to bear in mind that this is an unbalanced dataset. If a classifier/EWS picked non-transitioning every time, it would have a very high correct prediction probability of $209/244=0.86$. Therefore I don't think that correct prediction probability (which is used in the results section line 190-204) is a good measure of performance. Something like the F1-score may be more appropriate, which strikes a balance between the true positive rate and the true negative rate. ROC curves are also a great way to get performance

measures on a binary classification task (using the area under the curve). I would like to see which EWS perform best with a metric that is more suitable for an unbalanced dataset.

- It is mentioned throughout the manuscript that multivariate EWS outperformed univariate EWS (abstract, line 124, line 251, line 306). Can the authors explain how this statement is supported by the results? Line 190 seems contradictory to this: “univariate EWS estimated using expanding windows displayed the highest average probability of correct classification”. To me, what seems most noteworthy is the fact that the expanding window improves performance regardless of whether you use uni/multivariate data. This may be due to large number of missing data points in the plankton time series, and the relatively small amount of pre-transition data for a given lake.

- The time series are split into ‘transitioning’ and ‘non-transitioning’. Where do the time series with a regime shifts but no critical transition go? I think into the ‘non-transitioning’ category, but this should be made more clear in the manuscript. If this is the case, when the authors compute EWS in these non-transitioning time series, are they including the section with the regime shift, or trimming the data to just before the regime shift? If the former, one would expect a spike in variance during the regime shift, which could trigger false positives in this analysis. I think it would make sense to also trim the regime shift (non-transitioning) time series so the regime shift is not included in the EWS computation.

Minor comments

- The Github repository is well organised and well documented. However, I didn’t find any indication as to what each data file represents? There are 4 files in the data directory. I suggest that the authors indicate this somewhere in a readme file.

- It’s not clear to me what the difference is between the scaled and unscaled weights of the ML classifier. Please explain this somewhere, and why it has such a large impact on the EWSNet predictions.

- Am I right in thinking that break points were computed using total plankton densities? Did the authors consider EWS in total plankton densities instead of individual densities? Given that there were many zeroes in the individual plankton time series, I’d be curious to know if EWS have higher performance on the aggregated time series.

- There is more recent relevant work on machine learning for EWS that could be cited on line 315:

Patel and Ott, Using machine learning to anticipate tipping points and extrapolate to post-tipping dynamics of non-stationary dynamical systems, Chaos 2023.

Dylewsky et al., Universal early warning signals of phase transitions in climate systems, Interface, 2023.

- Fig 1B: what data is being shown here? Real lake data?

- How is the scaled metric score in Figure 2 computed? I didn't see it in the methods - sorry if I missed it.

- Line 62: CSD does not increase as such, it is the phenomena of an increasing return time following perturbations.

- Abstract: "recently developed machine learning techniques". I think this is too broad, since only a single machine learning technique was tested. I think it should read "a recently developed machine learning technique".

Typographical changes/suggestions

- Line 60: "bifurcation theory which states that a...." → "bifurcation theory which describes how a..."

- Line 102: "machine learning exploits"

- Line 753: "machine learning is limited to univariate time series" → "machine learning is applied to univariate time series" (there are techniques to apply it to multivariate time series)

- Figure 2: x-axis "Explanatory variable"

(Thomas Bury)

Reviewer responses

Reviewer #1

General comments

In the article “Early warning signals are hampered by a lack of critical transitions in empirical lake data”, the authors use time series from 9 lakes and different trophic levels (zooplankton, phytoplankton) to test and compare the ability of various early-warning indicators to predict critical transitions. The tested EWSs belong to three groups: classical univariate indicators, multivariate indicators and machine learning-based classification. In addition, the performance of the indicators was evaluated in combination with different preprocessing approaches for detrending and deseasoning and different computational approaches (rolling window, expanding window).

I find the manuscript is of general interest since the value and applicability of EWSs are still under debate. The contribution of this manuscript to the debate lays particular in the inclusion of different trophic levels, true positive and true negative transitions, different groups of EWSs, and the systematic investigation of the influence of different preprocess and computational approaches on the performance of the EWSs.

However, in the present form, I also have several points of criticism regarding the manuscript.

1) Most importantly, this regards the definition and used approach to identify critical transitions, which uses only one necessary but not sufficient criterium. Thus the critical transitions are not uniquely identified here, and the proclaimed ground truth in itself thus is highly uncertain. It is difficult for most rapid regime shifts found in empirical time series to rigidly differentiate between "State thresholds" and "Driver-state hysteresis" (following the terminology used by Andersen et al., 2009, Ecological thresholds and regime shifts: approaches to identification) and, in fact, is impossible without the presence of several shifts (see, for example, Scheffer et al. 2003 Catastrophic regime shifts in ecosystems: linking theory to observation).

The reviewer raises a critical consideration regarding the differentiation between state thresholds and driver-state hysteresis and agree that in its original form, but we believe our threshold generalised additive model (TGAM) approach is capable of identifying hysteresis when fitted in the environmental state space due to the flexibility of GAM shapes. While a second/third shift may appear necessary, the overlap in GAM fits is indicative of when considered in combination with the time series trend and, following the reviewers' comments, bimodality in state, even if a second shift has not yet occurred. We have consequently added an additional figure (Figure 2) to visualise the hypothesised TGAM behaviour across various transition types following Scheffer and Carpenter (2003) and Bestelmeyer et al (2011).

Figure 2. Hypothesised behaviour of each possible system dynamics under three complementary analyses used to classify the fate of a time series. These analyses fulfil the criteria of Scheffer and Carpenter²⁸ and Bestelmeyer et al.²⁹ for identifying alternative stable states in empirical data through i) time series shifts, ii) a hysteresis response to the control parameter and iii) multimodal distributions. Analyses i) and ii) are performed using threshold generalised additive models (TGAMs) of plankton density against time and environmental driver respectively. Thresholds/breakpoints are only permitted to occur

between adjacent time points. Analysis iii) identifies unimodal vs bimodal distributions of plankton density across the entire time series. We expand these analyses over^{28,29} to identify other forms of transition and provide a qualitative description for each transitions expected behaviour in the three analyses. In the first two columns, thick lines represent mean TGAM fit with shaded regions the confidence interval. Dotted lines are discontinuities between breakpoints. In the third column, lines represent the density of observations.'

That's not to say our approach is infallible and so we have caveated our ground truthings as a 'best guess' given the available data and the need for generic EWS ability.

2) Further, the author's claim to identify the ubiquity of critical transition needs to be toned down considerably since, firstly, as stated above, the authors can not claim to have identified critical transitions uniquely. Secondly, in the face of an estimated 304 million lakes (sensus Downing et al. 2006), 9 lakes, not even covering all climate zones, lake types etc., are nowhere near showing the ubiquity or none ubiquity of critical transitions.

We agree and did not intend 'ubiquity' to mean 'universal' rather 'very common'. Most abrupt shifts are assumed to be critical transitions when practitioners apply EWSs to their use case and so we aimed to caveat that assumption. We have removed this statement regarding ubiquity and hope the resubmission better frames our findings from that perspective.

3) Therefore the authors should take care of a stringent definition early in the text of the core terminology around (rapid) regime shifts, bifurcation and critical slowing down. Preferably this definition is based on an accepted use in the community. I suggest, for instance, Boettinger et al., 2013, Early warning signals: the charted and uncharted territories. After such a definition, it is helpful for the reader and avoids misunderstandings to use this terminology consistently throughout the text.

This is a key point we neglected to sufficiently cover. Consequently, we have added an additional glossary in Box 1 with referenced definitions.

Box 1

Figure 1. Classification tree relating the primary transition types and system dynamics relevant to early warning signals (EWSs) and regime shift detection. Terms are colour coded by their expectation to exhibit critical slowing down (the phenomenon quantified by EWSs).

In its simplest form, a regime shift is 'the process whereby an ecosystem suddenly changes from one alternative stable state to another'²³ whereas critical transitions (A) are but one possible mechanism of regime shift which can only be inferred (though not confirmed) from observational data if a number of criteria are satisfied^{12,24–26}. These criteria include: 1) an abrupt shift in time series, 2) driven by positive feedback mechanisms, 3) in response to an incremental increase in control parameter, and 4) results in a multimodal distribution of state values (alternative stable states). Alternative regime shift mechanisms are possible (B) and will display multiple criteria required for identifying a critical transition but will not include all of them. These mechanisms include stepwise changes in state resulting from a stepwise change in the control parameter (exhibits criteria 1 and 4), or noise-induced transitions that occur despite no changes in control parameter (exhibits criteria 1, 2, and 4)¹.

Glossary

Regime shift: Sudden or abrupt shift to an alternative attractor resulting from the influence of an external control parameter/driver or by the system's internal dynamics. A regime shift may be associated with bifurcations (after crossing control parameter thresholds), step changes in state (in response to step changes in control parameter) or limit cycles (cyclic changes due to the system's internal dynamics). These abrupt shifts may also occur across different trophic levels^{8,78–80}.

Bifurcation: Gradual changes in a control parameter drive 'qualitative' change in the behaviour of an equilibrium point in a dynamical system following the transgression of a bifurcation/tipping point^{78,81}.

Tipping point: A threshold value at which a dynamical system undergoes a sudden shift from one stable state to another alternative stable state in response to small stochastic perturbations⁸².

Critical transition/catastrophic bifurcation: A sudden shift from one steady state of a dynamical system to an alternate state via a fold/saddle-node bifurcation (a first-order or discontinuous transition). A discontinuous response follows incremental change in the control parameter crossing a critical value/bifurcation point. All critical transitions are regime shifts but the reverse is not necessarily true^{8,9,82}.

Non-critical transition/bifurcation: A non-catastrophic transition (a second-order or continuous transition) can occur via a transcritical, pitchfork or Hopf bifurcation and may not involve alternate stable states. However, bifurcation points are still present within the system^{8,9}.

Smooth transition: Continuous response to the control parameter in the absence of bifurcation points⁹.

Critical slowing down: The phenomenon whereby the real part of the dominant eigenvalue of the system approaches to zero in the vicinity of a bifurcation point (and eventually goes to zero at the bifurcation point) while the return/recovery rate to equilibrium upon perturbation becomes increasingly slow. At the tipping or bifurcation point there is no chance of recovery^{10,81}.

References:

- Andersen, T., Carstensen, J., Hernández-García, E. & Duarte, C. M. Ecological thresholds and regime shifts: approaches to identification. *Trends in Ecology & Evolution* **24**, 49–57 (2009).
- Scheffer, M. & Carpenter, S. R. Catastrophic regime shifts in ecosystems: linking theory to observation. *Trends in Ecology & Evolution* **18**, 648–656 (2003).

3. Dakos, V., Carpenter, S. R., van Nes, E. H. & Scheffer, M. Resilience indicators: prospects and limitations for early warnings of regime shifts. *Philosophical Transactions of the Royal Society B: Biological Sciences* **370**, 20130263 (2015).
4. Boettiger, C., Ross, N. & Hastings, A. Early warning signals: the charted and uncharted territories. *Theoretical Ecology* **6**, 255–264 (2013).
5. Strogatz, S. H. *Nonlinear dynamics and chaos: with applications to physics, biology, chemistry, and engineering*. (CRC Press, 2015).
6. Scheffer, M. et al. Anticipating critical transitions. *Science* **338**, 344 LP – 348 (2012).
7. Kéfi, S., Dakos, V., Scheffer, M., Van Nes, E. H. & Rietkerk, M. Early warning signals also precede non-catastrophic transitions. *Oikos* **122**, 641–648 (2013).
8. Wissel, C. A universal law of the characteristic return time near thresholds. *Oecologia* **65**, 101–107 (1984).

This being said, this ambiguity in definitions is one that we believe is a current challenge to practical users of early warning signals, and so we have added an additional sentence in both the Introduction and Discussion to explore this.

Lines 73-78

‘Major concerns of EWSs include their focus on ‘transitioning’ systems²⁰, data pre-processing¹⁴, and, in our opinion, the assumption that critical transitions are common in natural systems²¹. Crucially, there is a difference in definition between ‘regime shifts’ and ‘critical transitions’ (Box 1), but the two are often assumed the same^{18,19,22}. This synonymity leads to confusion regarding whether most ecological regime shifts are critical transitions and, if not, are EWSs appropriate for generic regime shift assessment.’

Lines 266-283

‘Generic EWS predictive ability is limited as it appears some system specific knowledge is necessary. Namely, understanding the potential for a critical transition/multiple stable states and identifying a mechanistic driver of transition are particularly key. These are not new arguments^{1,20} but are worth reiterating to avoid the conflation of critical transitions with any form of abrupt change/regime shift. The TGAM used here, when combined with bimodality quantification, identifies the same regime shifts as previous studies^{49,53–55} but only some of these were classifiable as critical transitions. We suggest it may be prudent to consider that temporal dynamics of driver variables (e.g. nutrient concentration) are themselves non-linear⁵⁶ and display pulse events⁵⁷ which can influence the mechanisms of regime shifts. For example, an anomalous year may push a system away from equilibrium in to a long transient, as potentially occurred during 1985 in Windermere’s zooplankton (Figure 2), or a step change in environmental conditions can result in novel communities^{1,58}. Consequently,

critical transitions can occur earlier, later, or not at all, even if a regime shift occurs. This is compounded further as the disambiguation of critical transitions from other regime shifts mechanisms is extremely complicated in empirical data^{28,29,59}, and the classifications we have made here are ultimately a ‘best guess’ given the data availability. We therefore encourage empirical regime shift and EWS researchers to consider the mechanisms driving shifts^{1,28,29} to maximise their reliability and appropriateness, and to not solely use EWSs as evidence of approaching tipping points.’

4) Here it could be helpful to add an accompanying diagram showing how terms relate, e.g. in the style of Boettinger et al. Fig. 1. I acknowledge some attempts to provide definitions by the authors (e.g. L49 ff, L74 ff); however, the definitions are not stringent in themselves and are not consistently used throughout the text. To exemplify the above point, I list some of the plethora of terms appearing in this text: abrupt non-linear changes, regime shifts, critical transitions, type of transition, whole ecosystem change, different forms of abrupt change, bifurcation-theory, process-driven regime shifts, tipping points, critical point, positive feedback loops, critical slowing down, alternative stable state, pulse events, step changes, smooth transition, step change/transient.

We agree this is a sensible addition and resultantly we have replaced Figure 1 in this resubmission with Box1 which contains a decision tree depicting the relationships between the various terms, and the aforementioned separate glossary. We thank the reviewer for the elegant solution.

5) Regarding the deficits in the methods section, I would like a more detailed and differentiated description of the pre-processing steps carried out regarding the monthly and annual time series. In addition, it remains unclear how the breakpoints identified in the annual resolution are applied to the monthly time series. In this context, I am also missing a critical discussion of why regime shifts have been identified at the zooplankton and phytoplankton level rather than the genera level and how this may have affected the ability to detect regime shifts, particularly because regime shifts do not necessarily affect all species/genera with the same quality, as the authors themselves state.

We apologise for the lack of clarity involving the pre-processing steps as we had believed our reporting in the Materials and Methods was sufficient. We have expanded the description on lines 447-465:

'Prior to early warning signal (EWS) assessment, each plankton time series was pre-processed via detrending and deseasoning using a range of techniques. Detrending is considered important for improving the reliability of EWS assessments⁵¹, so we applied three commonly used methods (linear detrending, LOESS smoothing, and gaussian smoothing) and compared assessments made to those based upon the raw time series. Linear detrending fits a linear model between time and plankton density, with the residuals of this model representing the detrended time series⁷⁹. LOESS, or local polynomial regression smoothing, subtracts a smooth curve fitted by local polynomial regression of span 0.5 from the raw time series³⁷, while gaussian kernel smoothing applies a linear filter, by subtracting the weighted moving average from the raw time series⁷⁹. Additionally, monthly time series were deseasoned as monthly plankton data is inherently seasonal⁸⁰, and the repeated non-linear cycles can hinder EWS capability⁸¹. We therefore applied three deseasoning techniques (averaging, additive decomposition, and STL) factorially with the detrending methods to identify the optimal combination. Averaging simply subtracts the average value for a given month from the current data point of that month⁷⁰, additive decomposition estimates the seasonal cycle from moving averages which is then subtracted from the raw time series⁸², and STL (seasonal trend estimation using loess) which also estimates the average seasonal cycle but uses local polynomials rather than linear/moving averages⁸³. All data pre-processing was performed using the EWSmethoDs R package v1.1.2⁶⁸.'

Our primary focus in this research was the loss of functioning/change of system state - a typical management aim for freshwater bodies, particularly those used as water resources. Consequently, we are interested in regime shifts at the system/community level. As this is the case, we focussed our classifications at the trophic level as, typically, ecosystem modelling compartmentalises observed communities into functional groupings, most notably trophic levels (Cox, P., Betts, R., Collins, M. *et al.* Amazonian forest dieback under climate-carbon cycle projections for the 21st century. *Theor Appl Climatol* **78**, 137–156 (2004); Christensen, V., and Walters, C. J. (2004). Ecopath with Ecosim: methods, capabilities and limitations. *Ecol. Model.* 172, 109–139; A. C. Patterson, A. G. Strang, K. C. Abbott, When

and where we can expect to see early warning signals in multispecies systems approaching tipping points: Insights from theory. *Am. Nat.* **198**, E12–E26 (2021)).

This is also true for alternative stable state research simulating aquatic dynamics pivotal to our understanding of critical transitions and their prediction (e.g Scheffer, M., Rinaldi, S., Kuznetsov, Y. A., & van Nes, E. H. (1997). Seasonal Dynamics of Daphnia and Algae Explained as a Periodically Forced Predator-Prey System. *Oikos*, *80*(3), 519–532; Priester, C. R., Melbourne-Thomas, J., Klocker, A. & Corney, S. Abrupt transitions in dynamics of a NPZD model across Southern Ocean fronts. *Ecol. Model.* **359**, 372–382 (2017)).

Trophic levels have been shown to strongly link functional groups (Brooks, D. R., Storkey, J., Clark, S. J., Firbank, L. G., Petit, S., & Woiwod, I. P. (2012). Trophic links between functional groups of arable plants and beetles are stable at a national scale. *Journal of Animal Ecology*, *81*(1), 4–13.; D’Alelio, D., Libralato, S., Wyatt, T. *et al.* Ecological-network models link diversity, structure and function in the plankton food-web. *Sci Rep* **6**, 21806 (2016)) and we therefore focussed our classification on trophic levels to be consistent with the historic literature and to build upon our general reconciliation of functioning with functional groups, or in their simplest form, trophic levels.

A second, practical consideration, was that we wanted to compare multivariate early warning signals to univariate signals. Therefore, trophic levels represent the lowest hierarchical classification valid across lakes that yield similar sample sizes across classes whilst maintaining the multiple time series needed for the multivariate approaches.

Lines 410-422

‘We also performed our classification procedure independently across phytoplankton and zooplankton trophic levels as this dataset gives us the opportunity to question whether critical transitions are shared across both components of the system. For example, it is plausible that a critical transition in one trophic level will not necessarily be matched by a critical transition in the other if the former trophic level is a driver of the second; a critical transition in one system component drives a non-bifurcation regime shift in another. We did not perform classification at lower taxonomic or functional levels as, ultimately, we are interested in the prediction of regime shifts at the system/community level. For ecosystem

managers, it is changes in functioning that is of concern ^{21,22} and it is the aggregate effect of system components that drive functioning ⁷⁴. Trophic levels also represent the simplest linkage between functional groups and is the typical method of compartmentalising ecosystem models ^{26,75}. Classifying at the trophic level therefore is consistent with the historic literature and builds upon our general reconciliation of functioning with functional groups, or in their simplest form, trophic levels.’

6) Moreover, although testing the expanding windows approach is an important aspect of the manuscript, details on the method remain unclear, even after following the given citations.

Expanding windows estimate the indicator value for a given time point and standardise it by the running mean and running standard deviation of all previous indicator estimates using the equation:

$$EWS_t = \frac{ews_{1:t} - \overline{EWS_{1:t-1}}}{sd(EWS_{1:t-1})}$$

where t is the current time point, ews is the estimated early warning signal indicator value, and EWS is the expanding window/standardised running EWS value. The method therefore iteratively updates the estimated EWS value with each successive time point, with a ‘warning’ signalled when the EWS value exceeds either 1 or 2 standard deviations from its running mean. Often a ‘burn in’ period is applied so that a sufficient running mean and standard deviation is achieved prior to assessment. We have added this description to the manuscript on lines 486-495.

‘The alternative expanding window computation incrementally introduces new data after a set burn in period. Each indicator is standardised by subtracting its running mean from its calculated value at time t before division by its running standard deviation ³³ using the equation:

$$EWS_t = \frac{ews_t - \overline{EWS_{1:t}}}{sd(EWS_{1:t})}$$

where t is the current time point, ews is the estimated early warning signal indicator value across all data up to t , and EWS is the standardised running EWS value of all previous estimates. A composite metric can then be constructed by summing all individual indicator

values calculated per t . An oncoming transition is consequently identified when the indicator/composite metric exceeded its expanding mean by a certain threshold value. Here, we set that threshold at two standard deviations due its favourable performance relative to alternative threshold values ⁶⁴,

7) Most importantly, details are missing on how the authors derive the central success/failure data table, which would be necessary for reproducibility.

We apologise for this ambiguity. Although both rolling windows and the machine learning model generate bounded numerical data, the expanding windows are designed to only be interpreted as signal-no signal (Drake and Griffen, 2010; Clements and Ozgul, 2016). As the reviewer highlights, the indicators would not be comparable in this form, hence the conversion of all indicators to success-failure. A success was identified if a signal was identified in a lake classified as being a critical transition, or when no signal was identified in all other circumstances.

This information is available on lines 521-530:

‘Additionally, as the various EWS method classes all generate different outputs, we converted these outputs into the binary presence-absence of a ‘warning’ (Figure 4C). For rolling window computations, a warning was accepted if a positive Kendall tau correlation was in the 95th quartile of Kendall tau correlations from a dataset permuted from the original time series ¹¹, for expanding windows when the two standard deviation threshold was exceeded for two or more time points ⁷⁴, and for EWSNet, when the model predicted a critical transition (i.e. the strongest probability)⁶¹. This presence-absence of a warning was then compared to the ground-truth labels identified by the TGAMs, where a ‘warning’ signalled in a system classified as a critical transition, or the absence of warning in system not classified as critical transition, was considered a success, and resulted in a binomial dataset of successes and failures. We only considered critical transitions here as these represent the primary classification of concern, due to their abrupt and hysteretic nature.’

8) In particular, it is unclear how results from univariate EWS indicators, with one result per genus level, were made comparable to results from multivariate EWS indicators, which return one result for multiple time series (system).

The complication of comparing univariate to multivariate was a particular challenge of this study and led us to consider alternative options to the suite of methods typically used in binary classification problems and those suggested by Reviewer #3 (namely F1 statistic and Area Under Curve). We settled on using Bayesian logistic regression as the method allows us to control for various interdependencies and potentially confounding structures in the dataset including the unbalanced transitioning vs non-transitioning sample sizes, repeated EWS assessments within a lake, and the difference in sample sizes between univariate and multivariate indicators.

Our specific model achieves this by weighting each data point against the number of ‘trials’ or assessments made by that indicator (under the binomial distribution), and allows the error structure of each indicator estimate to vary between lakes but correlate within a lake using random effects (Bolker et al. 2009; <https://www.sciencedirect.com/science/article/pii/S0169534709000196>). And critically, additional random effects are fitted using the ‘ground-truth’ labels to minimise the bias towards Type I errors potentially driven by a larger pool of non-transitioning lakes relative to transitioning. We have clarified this point on lines 581-589:

‘During interpretation, we back-transformed the log odds into probabilities of correct classification, and used the overlap of the posterior distribution’s credible intervals against 50% to identify EWS approaches that provide better estimates than chance. Modelling the probability of correct classification in this way allows us to control for confounding factors in the dataset, namely lake identity and the varying number of trials (i.e. EWS assessments) between lakes and EWS methods. This control is not possible using the F1-statistics and receiver operator curves typically used for binary classification tasks^{36,37} which are limited to weighting based upon unequal sample sizes and cannot estimate codependencies between the repeated measurements inherent to ecological data.’

And finally, we further ensured comparability between univariate and multivariate assessments by framing our discussion and comparisons at the trophic and system level as ultimately the goal of EWS indicators is to provide system level predictions using a representative state variable; mathematical models with alternative stable states resulting from a fold bifurcation are typically characterised by a single variable (e.g. Scheffer, 1990;

Lade and Gross, 2012) and in planktonic lake systems, it is often whole system change that is of managerial interest rather than specific species.

9) L27: “... often inaccurate...” This is a strong claim since most EWSs have a solid theoretical grounding. So the question would be: inaccurate in which regard?

We have modified this sentence to better indicate our belief that EWS inaccuracies have primarily been found when applied to real-world data rather than simulated (where the strong theoretical grounding can be idealised). Line 26

‘This has led to the development of a suite of early warning signals (EWSs), that unfortunately often perform inaccurately when applied to real-world observational data.’

10) L31: From theory, it is clear that EWSs derived from CSD are not expected to predict all types of (rapid) regime shifts. Already here, I found the fuzzy use of terminology confusing.

We hope that the updated text and additional glossary will minimise this ambiguity (Box 1).

11) L32,L35: “... identify the type of transitioning ..”, “.... different forms of abrupt change...”. Be more precise. What transitions (group of transitions) do you distinguish (critical transition vs no critical transition, Table S1)

The abstract has been changed extensively with this section now reading:

‘...Most of this work has built on the theory of bifurcations, with the assumption that critical transitions/catastrophic bifurcations are common features of complex ecological systems. This has led to the development of a suite of early warning signals (EWSs), which unfortunately perform inaccurately in observational data. Consequently, techniques have been proposed to overcome EWS limitations by analysing multivariate time series or applying machine learning. It however remains unclear whether critical transitions are the dominant mechanism of regime shifts and – if they are present – whether classic and second-generation EWS methods predict them...’

12) L37-38: Given that which group of EWS indicators performs best seems highly dependent on the temporal resolution of the time series and the chosen preprocessing and calculation approach, I don't see how you arrive at this general summary/conclusion.

Following reanalysis in response to all reviewers to make explicit the compensation for unbalanced sample sizes, we have rewritten this concluding sentence:

'We find few instances of critical transitions in our lake dataset, with different trophic levels often expressing different forms of abrupt change. The ability to predict this change is highly processing dependant, with most indicators not performing better than chance, multivariate EWSs being weakly superior to univariate, and a recent machine learning model performing poorly.'

13) L40: "... predict change ..." This is very generic - better "abrupt regime shifts" (see general comments regarding terminology).

Our suggestion to focus on general resilience loss/system change has been edited in the abstract (which has strict word limits). We believe that focussing on general/generic ecosystem resilience changes are more meaningful than a sole focus on regime shifts and/or critical transitions. That's not to say that pre-emption regime shifts is not important, we stress that it is, but resilience measures (such as those suggested by reviewer #2) have wider utility outside of regime shifts. Line 37-40

'Our results suggest that predictive ecology should start to move away from the concept of critical transitions and develop methods suitable for predicting resilience loss in the absence of the strict bounds of bifurcation theory.'

14) L76-78: No, this is a necessary but not sufficient criterium.

We hope that our new Box 1 better defines the necessary criteria of Scheffer and Carpenter (2003). This sentence has resultantly been removed with all focus now on Box 1.

15) L84: Be more precise. Which ones?

This sentence is now contextualised by the sentence on line 117-127:

‘In this work we classify regime shifts, critical transitions, non-critical transitions and stationary systems (Figure 2) in nine long-term lake monitoring datasets (Figure 3)... The precise classification of lake fate reveals that many accepted regime shifts are not critical transitions, with different trophic levels responding uniquely to environmental change.’

16) L89: “uses” instead of “users”

Changed accordingly.

17) L94: I had trouble finding this statement in the cited article; however, if this is the case, this also leads to uncertainty in your transition type classification, which needs to be critically discussed.

The citation refers to the second clause of the sentence regarding issues focussing just on transitioning systems. We have now provided a citation for the first clause (Rowland, J.A., Nicholson, E., Murray, N.J., Keith, D.A., Lester, R.E. and Bland, L.M. (2018), Selecting and applying indicators of ecosystem collapse for risk assessments. *Conservation Biology*, 32: 1233-1245. <https://doi.org/10.1111/cobi.13107>) and have rewritten this sentence as follows on line xx:

‘Reliance on univariate time series can therefore makes it challenging to define an ecosystem’s dynamics as stable or transitioning³³, particularly as most previous work neglects stationary time series²⁰,

18) L123-125: Given that you are investigating only 9 lakes and your applied classification method can not uniquely determine a critical transition, it is necessary to tone down this statement.

The statement has been altered accordingly (lines 250-255) and now reads:

‘Motivated by debates surrounding multiple stable states in ecology and the need for reliable and generic critical transition detection tools, we assessed the prevalence of critical transitions in a range of empirical lake systems. We then compared the ability of current early warning signal (EWS) methods to correctly predict ecosystem fate regardless of transition type or trophic level. We found that multiple regime shifts were identifiable across our lake network, but only a proportion of these were critical transitions.’

19) L 125-126 Given the versatility of your results regarding preprocessing, temporal resolution, true positives, and true negatives, I would strongly recommend refraining from such generic, off-hand summaries.

We have followed the reviewer’s suggestion throughout. We hope the changes are sufficient.

20) L382: How do downstream deseasoning and detrending apply to yearly data?

Detrending applies to yearly data but deseasoning does not. We therefore applied the optimum detrending identified from monthly time series to the yearly data and have clarified this statement on lines 447-448:

‘Prior to early warning signal (EWS) assessment, each plankton time series was pre-processed via detrending with monthly time series further deseasoned using a range of techniques.’

21) L383-384: The disappearance of a species can be seen as an important change in community composition and can possibly also depict an abrupt change. How does the technical need to exclude those time series bias your data against (sudden) changes? Please critically discuss how some of your data processing steps could have affected your conclusions.

The reviewer is correct that the loss of taxa is an important contributor to both state change and possible driver/symptom of abrupt change. However, we do not believe that our data processing regarding time series selection will dramatically influence our ability to test early warning signal (EWS) ability. Genera were only excluded after the TGAM system

classification (to minimise the same concerns as the reviewer) which was clear in the original submission. This has been rectified by rearranging the Materials and Methods so that data processing is explained after the TGAM models are introduced.

We were, however, forced to drop genera for EWS assessment as deseasoning will otherwise introduce spurious cycles over periods of persistent zeroes, and certain multivariate EWSs (maxCOV, mafSD, mafAR, pcaSD, pcaAR) are unable to form stable covariance matrices between those species. Therefore, to maximise indicator usage, we elected to exclude those genera. We have now added a discussion of this influence on lines 303-321.

“With real world time series, there is no ideal data for EWSs due to their typically high variability and cyclical nature driving false positive warnings⁶. Detrending and deseasoning are therefore necessary but are no silver bullet for accurate EWS assessments. Deseasoning is particularly complicated and capable of introducing spurious signals when the time series’ value (i.e. plankton density/abundance here) are persistently close to zero. The sensitivity of EWS ability to data pre-processing reported here is therefore not unexpected^{51,61} and arguably weakens EWS practicality. Choosing what ‘system’ EWSs are quantifying is also a key determinant of EWS ability. Here we focus on the entire lake ecosystem rather than specific populations of interest (as occurs during fishery management^{22,62}) but extrapolate that univariate signals from individual plankton genera as representative of trophic/lake level regime shifts. This is typical EWS usage^{18,22,32} although we know different species/genera vary in their expression of critical slowing down^{30,32}. Such taxon specific EWS behaviour limits the relevance of many genera in our lake dataset and influences our null findings from naïvely performing EWS assessments when no information is available to select specific taxa. Unfortunately, this is the approach many managers are required to take in the absence of calibrated lake or ecosystem models and so we urge caution when choosing where to apply EWSs. Trophic and functional groupings appears valid levels to classify regime shifts, but EWSs require linear stability analysis to identify representative taxa. Multivariate EWSs mitigate some of this requirement but were not as successful as previously reported in simulated data^{35,38}.”

22) L385-386: Add: “The two final datasets on a monthly and yearly temporal resolution for each lake

Added accordingly.

23) L390-391: How does this statement relate to yearly data?

Only monthly data were deseasoned while both yearly and monthly were detrended. This sentence has been amended to clarify this. Lines 447-448

‘Prior to early warning signal (EWS) assessment, each plankton time series was pre-processed via detrending with monthly time series further deseasoning using a range of techniques.’

24) L410-412: This is a necessary but not sufficient criterium!

We agree and have responded to this comment in reply to comment 3).

25) L414, L154: If you identify a transition on yearly resolution, how did you divide the monthly time series into pre-and post-transitioning phases?

Monthly time series were partitioned under the year identified from the yearly data. We repeated the classification process using the monthly data which was in near agreement with the yearly (although many more breakpoints were identified and bimodality was never detected - presumably due to seasonal effects smoothing the kernel density estimate). Transition dates were often within the year identified by the yearly and consequently, we subset time series prior to that year for comparability while also being conservative against the higher uncertainty of breakpoints estimated from monthly data compared to yearly data..

The equivalent summary table to Table S2 for monthly data is provided below. Monthly dates are reported numerically where the difference between months is 1/12 or 0.083: 1970.000 is January 1970, 1971.083 is February 1970, 1971.500 is June 1971 etc. This is required to program breakpoints in the TGAM time series fit.

Lake	Trophic level	Explanatory variable	Breakpoint date	Bimodality detected
Kasumigaura	Phytoplankton	Time	NA	No
		Environment	1998.667	

	Zooplankton	Time	2009.500	No
		Environment	2009.500	
Kinneret	Phytoplankton	Time	1994.917	No
		Environment	1994.917	
	Zooplankton	Time	1995.250	No
		Environment	1985.167	
Loch Leven	Phytoplankton	Time	1998.917	No
		Environment	NA	
	Zooplankton	Time	1996.250	No
		Environment	NA	
Lower Zurich	Phytoplankton	Time	2001.000	No
		Environment	1986.750	
	Zooplankton	Time	NA	No
		Environment	NA	
Mendota	Phytoplankton	Time	NA	No
		Environment	2010.083	
	Zooplankton	Time	NA	No
		Environment	2011.000	
Monona	Phytoplankton	Time	2012.417	No
		Environment	2007.000	
	Zooplankton	Time	2011.417	No
		Environment	2011.250	
Upper Zurich	Phytoplankton	Time	1993.500	No
		Environment	1993.000	
	Zooplankton	Time	NA	No
		Environment	NA	
Washington	Phytoplankton	Time	1970.667	No
		Environment	1970.667	
	Zooplankton	Time	NA	No
		Environment	1986.417	
Windermere	Phytoplankton	Time	1988.083	No
		Environment	NA	
	Zooplankton	Time	1986.250	No
		Environment	NA	

26) L416: How were plankton densities derived from the genus-level time series? As sums? How could this affect your ability to detect an abrupt shift (critical or otherwise), e.g. due to portfolio effects and not all genera might be affected equally by a regime shift? Also, here a critical discussion would be required.

Plankton genera densities were calculated as sums of individual species (the maximum number of species pooled to a single genus was five). As the reviewer highlights, this may influence EWS capability (though not the TGAM pre-classification as classification took place at higher hierarchical organisation), decreasing EWS detection due to buffering effects. We conversely focussed on genera level to minimise false positives arising from high zero time series, and a belief that the high functional similarity within plankton groups – i.e. Hutchinson’s paradox (Hutchinson, G. E. (1961). The Paradox of the Plankton. *The American Naturalist*, 95(882), 137–145.) – will respond similarly to stress. It is understood that there is a phylogenetic signal for the likelihood of extinction in plants (R. Dinnage, A. Skeels, M. Cardillo, Spatiophylogenetic modelling of extinction risk reveals evolutionary distinctiveness and brief flowering period as threats in a hotspot plant genus. *Proc. R. Soc. B Biol. Sci.* **287**, 20192817 (2020)) and vertebrates (Capdevila, P., Noviello, N., McRae, L., Freeman, R. & Clements, C.F. (2022) Global patterns of resilience decline in vertebrate populations. *Ecology Letters*, 25, 240–251), which when combined with the general

functional group approach taken in plankton research, implies grouping the genera level (a significantly lower hierarchical level than the typical Phylum level used) is unlikely to mask true EWSs. This justification is given on lines 439-445:

‘Plankton species were then pooled to genus level to minimise the likelihood of zero densities, with a genus further dropped if they disappeared for a period longer than 12 months. This is necessary as downstream deseasoning can spuriously introduce cycles of non-zero densities into periods of zeroes. In addition, the strong phylogenetic signal in vertebrate and plant abundance trends implies that aggregating to genera level from species is unlikely to mask the strong CSD in critically transitioning taxa. The two final datasets of monthly and yearly temporal resolution for each lake consequently consisted of genus level densities across the two plankton trophic levels.’

and further discussed on lines 309-321:

‘Choosing what ‘system’ EWSs are quantifying is also a key determinant of EWS ability. Here we focus on the entire lake ecosystem rather than specific populations of interest (as occurs during fishery management^{22,63}) but extrapolate that univariate signals from individual plankton genera as representative of trophic/lake level regime shifts. This is typical EWS usage^{18,22,33} although we know different species/genera vary in their expression of critical slowing down^{30,33}. Such taxon specific EWS behaviour limits the relevance of many genera in our lake dataset and influences our null findings from naïvely performing EWS assessments when no information is available to select specific taxa. Trophic and functional groupings appear valid levels to classify regime shifts, but EWSs require linear stability analysis (LSA) to identify representative taxa. Unfortunately, many managers are unable to take this the approach due to the lack of calibrated lake/ecosystem models appropriate for LSA. We therefore urge caution when choosing where to apply EWSs. Multivariate EWSs mitigate some of the requirement for LSA but were not as successful as previously reported in simulated data^{36,39}.’

27) L450-451: Over which group of time series were this averaged, zooplankton vs phytoplankton, i.e. trophic level? What is, in your case, the “system”?

Averages and dimension reductions were made over a trophic level to match the classifications made by TGAMs and bimodality coefficients. We are interested in overall lake functioning shifts (this is what we consider the ‘system’) and use the different genera and trophic levels as sub-samples of the lake system to balance between the simplified data structures EWSs require, and the inherent complexity of natural lake ecosystems. This has been clarified on lines 471-475.

‘Multivariate EWSs expand these assessments from single time series to multiple by either averaging across univariate EWSs or by extracting CSD information from a dimension reduction of the system (Figure 4B). Here we averaged performed dimension reductions across all time series within a trophic level to match the classifications made in the Critical transition pre-classification section.’

28) L463-464: The calculation stays unclear, despite also following the cited articles since it is unclear how, e.g. autocorrelation at time t is calculated.

We hope our response to comment 6) clarifies this question.

29) L473: Define "smooth transition". How did you map this to your “ground truth” (Table S1) for the performance evaluation?

Smooth transition is now defined in Box 1, though please note, that EWSNet’s authors define smooth transitions as we define non-critical bifurcations. Certain EWSNet authors are co-authors on this manuscript and agree with our Box 1 definitions as EWSNet’s Smooth Transitions are trained upon transcritical, pitchfork and Hopf bifurcations, not the gradual smooth dynamics explored by Kefi et al (Kéfi, S., Dakos, V., Scheffer, M., Van Nes, E.H. and Rietkerk, M. (2013), Early warning signals also precede non-catastrophic transitions. *Oikos*, 122: 641-648).

We have rewritten this sentence accordingly. Lines 499-502:

‘EWSNet utilises the entirety of the pre-transition time series to provide probabilities of the likelihood of 1) a critical transition, 2) a smooth transition (please note this class is analogous to non-critical transitions defined in Box 1), or 3) no transition.’

30) L494: To follow this section, I would find it helpful if you would refer to the corresponding tables in the supplementary material.

Corresponding tables have been referenced throughout the section: Early warning signal ability including on lines 533-536:

'To estimate the classification ability of each EWS method, we developed a series of Bayesian hierarchical models using success/failure as response variable. Early warning signal method class and the specific EWS indicator itself were explored as categorical fixed effects in separate models: EWS method class (Table S8-S9) and indicator (Table S10-S13) ability.'

31) L484-485, L491-L492: How exactly was it handled that for the univariate measures, there is one success/ failure result for every genius level time series, while for multivariate methods, which are calculated across multiple time series representing "the system" (Table S2), there is only one.

The binomial multilevel modelling approach inherently controls for the discrepancy in sample sizes between univariate and multivariate by modelling not success-failure, but the total number of successes across the number of trials. Here, as the reviewer identifies, trials differs between multivariate and univariate, but the model resultantly weights based upon the total number of trials within an indicator/computation method class and allows the two forms of measures to be comparable. We answered this concern in more detail in response to point 6).

32) L507: 15 levels.

We apologise as we are unsure what the reviewer is referring to here. If regarding 'Early warning signal method class', then five levels are possible (*univariate rolling, univariate expanding, univariate machine learning, multivariate rolling, and multivariate expanding*). If EWS indicator, then 21 are possible (*ar1, SD, skew, ar1 + SD, ar1 + skew, SD + skew, ar1 + SD + skew, meanAR, maxAR, meanSD, maxSD, eigenMAF, mafAR, mafSD, pcaAR, pcsaSD, eigenCOV, maxCOV, mutINFO, scaled EWSNet and unscaled EWSNet*). If regarding detrending and deseasoning combinations, there are four categories for each (*detrending =*

none, linear, gaussian, LOESS; deseasoning = none, average, decomposition, STL) and resulting 16 combinations (4*4). We apologise again for misunderstanding and hope this explanation is sufficient clarification.

33) L516-518: I can not fully reconcile this information to the one given in L171-172, particularly because also in tables S3-S7 there seems to be no factor level “none-none”, even so in the table headers “assessments made on raw data” is mentioned.

Each estimate reported in table S3-S7 are coefficient differences (i.e. improvement) compared to factor level “none-none”. The raw time series (none-none factor level) coefficient has not been reported as its coefficient is represented as an intercept (i.e. the mean ability) rather than a coefficient difference. This is the standard when dummy coding categorical variables in linear regression models. We therefore felt it would confuse readers as all other coefficients are relative differences. While we are also ultimately interested in the improvement of each detrending and deseasoning combination over none-none, rather than the mean estimates themselves as we explore this question in Figures 5 and 6. We have therefore added the following statement to the caption of each table to clarify this:

‘Each estimate is therefore the relative improvement of that factor level versus the none-none data pre-processing’

and altered the estimate column heading to read:

‘Estimated improvement against none-none data pre-processing (median)’

If, however, the editor and reviewer believe the none-none intercept should be reported we are happy to make this amendment. Table S3 would therefore look as below if none-none factor level is included:

Pre-processing combination (detrending method – deseasoning method)	Median coefficient	Lower 95% credible interval	Upper 95% credible interval	Rhat	Effective sample size
none-none	0.076	-2.313	2.469	1	6065.53
linear-none	0.022	-0.281	0.326	1	4404.74
loess-none	-0.029	-0.336	0.278	1	4645.2
gaussian-none	-0.04	-0.349	0.271	1	4485.97
none-average	0.022	-0.287	0.338	1	4698.69
none-decomposition	0.05	-0.256	0.368	1	4436.14
none-stl	-0.018	-0.32	0.295	1	4483.88
linear-average	0.009	-0.299	0.311	1	4497.63
loess-average	0.035	-0.274	0.341	1	4496.95
gaussian-average	0.023	-0.282	0.339	1	4616.81
linear-decomposition	0.037	-0.273	0.345	1	4429.23
loess- decomposition	0.009	-0.296	0.315	1	4933.83
gaussian- decomposition	0.011	-0.298	0.324	1	4567.6
linear-stl	0.079	-0.223	0.39	1	4381.49
loess-stl	0.025	-0.282	0.336	1	4682.63
gaussian-stl	0.041	-0.271	0.346	1	4460.24

34) A table of content would be helpful to navigate the material and get an overview of the various tests.

We believe the reviewer is referring to the supplementary and so have added a table of contents to its title page. We apologise if this is not what was requested.

‘Contents

Figure S1: Lake regime shift classification using threshold generalised additive models and kernel densities.

Figure S2-S7: Posterior parameter estimates and diagnostic chain trace plots for Bayesian models.

Figure S8-S10: Diagnostic posterior predictive checks for Bayesian models.

Table S1: Lake breakpoints and bimodality identified by Figure S1.

Table S2: Description of early warning signal indicators used.

Table S3-S7: Model summary tables for the estimated improvement of data pre-processing on early warning signal ability in critically transitioning time series.

Table S8-S9: Model summary tables for early warning signal computation technique ability (i.e. univariate vs multivariate, rolling vs expanding windows) in yearly and monthly data.

Table S10-S11: Model summary tables for individual early warning signal indicator ability (in yearly and monthly data) in predicting critical transitions.

Table S12-S13: Model summary tables for individual early warning signal indicator ability (in yearly and monthly data) in predicting non-critically transitioning time series.'

35) What is the temporal resolution of the time series for the results in tables S3-S7.

Tables S3-S7 report monthly detrending and deseasoning, as deseasoning is only relevant for monthly time series.

36) L130 A map depicting the location of the lake would be appropriate.

As requested, we have added an additional map figure with both locations and area of the lake network (Figure 3).

37) In addition, a table summarizing under which temporal resolution and type of grouping (overall, true positive, true negative) which group of EWS + preprocessing performed best would be helpful for the reader to keep track of the specific results.

The reviewer makes a good suggestion and we have added a table (Table 2) reporting this information.

Table 2. Optimal detrending and deseasoning combinations for each early warning signal computation method across time series resolutions.

Data resolution	Early warning signal computation method	Optimal detrending method	Optimal deseasoning method
Monthly	univariate rolling window	linear	Seasonal and Trend decomposition using Loess (STL)
	univariate expanding window	linear	decomposition
	multivariate rolling window	gaussian	None

	multivariate expanding window	gaussian	averaging
	EWSNet (univariate machine learning model)	gaussian	None
Yearly	univariate rolling window	linear	NA
	univariate expanding window	linear	NA
	multivariate rolling window	gaussian	NA
	multivariate expanding window	gaussian	NA
	EWSNet (univariate machine learning model)	gaussian	NA

38) L145-147, L150-151: As mentioned before, this is a necessary but not a sufficient criterium.

We have now altered the phrasing here and directed readers towards Box 1 and the new Figure 2 for the hypothesised behaviour of different system dynamics under our TGAM/bimodality approach. Lines 143-149

‘Using threshold generalised additive models (TGAMs), we identified optimal break points in each lake’s total phytoplankton and total zooplankton density through both time and the environmental ‘state-space’, and quantified bimodality in extension of the approaches of Scheffer and Carpenter²⁸ and Bestelmeyer et al.²⁹ (see Materials and Methods)’

‘When comparing estimated break points between the time series and environmental models with state bimodality...’

39) L185: Add reference to table S8, i.e. “ cross-method comparisons (table S8) ...”.

Corrected accordingly.

40) L184-185: How does this apply to the yearly resolution time series?

Deseasoning does not apply to yearly time series and we have clarified this on lines 447-448:

‘Prior to early warning signal (EWS) assessment, each plankton time series was pre-processed via detrending with monthly time series further deseasoned using a range of techniques.’

41) L192: "...tables S8-S13".

Corrected accordingly.

42) L208: According to Figure 3, it should be “decreased”.

Here we intended ‘increasing data resolution’ to be higher frequency of measurements i.e. monthly data. However, we realise this may be confusing as increasing resolution could be interpreted both ways and so have rewritten this sentence (line 208):

‘Univariate and multivariate rolling window EWSs especially declined in robustness when applied to yearly data relative to monthly.’

43) L225-231: Refer to the appropriate tables in the supplementary in this section.

We had initially not referred to the supplementary tables in this section as they had been introduced in the previous. It should also be noted that the reported estimates will differ to the raw coefficients presented in the supplementary tables as we have back transformed from odd ratios back to probability for interpretability (stated on lines 192-194)

‘From this section onwards, binomial model estimates have been inverse-logit transformed from log odds to probabilities to improve interpretability. For raw model estimates please refer to Tables S8-S13, and Figures S2-S10 for model diagnostics.’

Similarly, certain reported values here are calculated across models to give an overall representation of ability, rather than being directly quoted from the supplementary tables. We have provided the R script to calculate these summary statistics in the linked Zenodo repository – script name = “*descriptive_results.R*”.

We have referred to Supplementary Tables where appropriate. Lines 239-243.

‘Autocorrelation at lag-1 (ar_1) was the most reliable rolling window univariate EWS for critical transitions (Tables S10-S11). Multivariate rolling window indicators such as mean

autocorrelation (meanAR), PCA variance (pcaSD) and the dominant eigenvalue of the maximum autocorrelation factor dimension reduction (eigenMAF) were particularly effective in transitioning time series but weak in non-transitioning data (Tables S10-S13).'

44) L233: Specify “this”.

Changed accordingly to suggest that the computation level interpretation of EWS is unhelpful and so interpreting at the indicator level is more informative. Lines 234-235

'This lack of coherence therefore suggests that individual indicators are highly variable and so should be considered individually.'

45) L234: Specify “the method”.

This phrase has been written and is no longer present. The new sentence reads:

Lines 234-239

'However, the dichotomy in prediction abilities observed for EWSNet was maintained across indicators and computation techniques. Composite univariate EWSs⁵² computed via expanding windows (e.g. $ar1 + SD$, $ar1 + SD + skew$) were reliable across resolutions in not critically transitioning time series but maintained ~ 0.5 ability in critically transitioning time series.'

46) The discussion in the present state should be shortened, and critical discussion to potential influences of methodological decisions added, particularly on the used “ground truth”, which was established on necessary but not sufficient criterium only.

The Discussion has extensively been shortened and an additional paragraph added on the influence of time series preprocessing, resolution of time series, and aggregation of time series to genera vs trophic level upon our results. Lines 447-448 – response to comment 21).

The ground truth labels have also been improved using the work of Scheffer and Carpenter (2003) and Bestelmeyer et al (2011) which we believe now fulfils many of the required

criterion for a critical transition (Box 1, Figure 2) and implies the presence of positive feedback loops which are challenging to identify exclusively from observational data.

47) L256-257: I do not see how you come to this clear conclusion, seeing the very mixed results. The performance and appropriate pre-processing seems situation dependent, e.g. temporal resolution of data; maybe a more differentiated recommendation would add value.

This statement has been edited to better synthesise this variability following all reviewers' suggestions.

48) L340-342: Citation with evidence in regard to this.

The citation(s) Ushio et al. (2018) and Medeiros et al (2022) are attached to the manuscript statement. Lines 339-345.

'While some of the measures tested here are considered stability indicators (e.g mutual information³⁵), more complicated measures have recently emerged independent from the assumption for local stability. For example, Ushio et al.⁶³ exploit empirical dynamic modelling to estimate the Jacobian matrix of a multivariate community and extract a stability index which accurately diagnoses vulnerable periods in fish communities. This has been developed further by Medeiros et al.⁵⁹ to identify key species for management based upon their contribution to the system's Jacobian and Grziwotz et al. for univariate time series⁶⁴

Reviewer #2

The paper by O'Brien et al "Early warning signals are hampered by a lack of critical transitions in empirical lake data", as stated by the title and abstract, discusses observed variables from nine lakes, and compares multiple early warning signals.

1) The paper obviously contains a lot of results, but when the authors claim that this or that indicator performs better or worse in terms of detecting transitions, this is based on statistical characteristics rather than on ground truth information about transitions – this can be caused by parameters of fitted models and pre-trained setups.

The reviewer raises a valid point that the ground truth labels used here are based upon observational time series, but we wish to clarify that the purpose of our analyses were to calssify using only the data available to lake/system managers. For our systems, four are known to have undergone a regime shift (Kasumigaura, Kinneret, Mendota, Washington) with one of those considered to have been a critical transition (Kasumigaura - Fukushima, T. & Arai, H. Regime shifts observed in Lake Kasumigaura, a large shallow lake in Japan: Analysis of a 40-year limnological record. *Lakes & Reservoirs: Research & Management* **20**, (2015)) based upon statistical characteristics such as segmented regression. We go further by applying the holistic suggestions of Scheffer and Carpenter (2003) and Bestelmeyer et al (2011) to disambiguate critical transitions/bifurcations from other forms of sudden shift while also identifying continuous but non-linear changes. Other than system specific modelling, we do not believe there is an observational way to identify forms of transition other than those proposed in the above references. That being said, we also believe, and suggest, that because of this, system specific modelling approaches likely represent the best future direction if managers can develop models appropriate for their system.

2) Furthermore, many of the indicators may contain issues by construction: for example, multivariate indicators aggregate variables in a simple way, and some of those may have opposite dynamics, while others may be cross-correlated and lead to a bias in estimates such Kendall's tau. This can affect the resulting multivariate indicators to such extent that any conclusion about its results may be misleading.

The reviewer is correct in their concern, and we do not disagree that such aggregation can weaken the power of such indicators in empirical data. However, we wish to clarify that we did not develop these aggregated variables and that the focus of the paper was to explore the diversity and efficacy of EWS methods currently available and accessible to managers of systems susceptible to critical transitions. We have therefore applied each EWS indicator relatively naively to assess whether even in the absence of targeted use (as in all previous empirical cross system comparisons– e.g. Gsell et al. 2016, Burthe et al. 2016), there is merit to these approaches, or whether there is a need for system specific indicators. We believe the latter is true.

3) I wonder if the authors are familiar with works of Killick and their R package “changept”? <https://cran.r-project.org/web/packages/changept/index.html> If the authors aim to distinguish abrupt transitions, this is a suitable tool for such analysis. Furthermore, I cannot see references to true multivariate analysis (i.e., not just aggregates or averages) developed by Williamson and Lenton (Chaos, 2015). For comprehensive comparison, the authors should include these methods, in my opinion

We thank the reviewer for their suggestion regarding the R package *changept*. The threshold generalised additive model approach used here is almost a direct analogue to the segmented regression model fitting provided by that package but allows non-linear trends as well as break points to be fitted. This consequently provides us the additional capability to distinguish one off anomalous years/transient dynamics that are not true regime shifts from abrupt and stable transitions.

With regard to the work of Williamson and Lenton (2015), we were not aware of it prior to the reviewer’s suggestion so we thank them for drawing it to our attention. Williamson and Lenton (2015)’s approach is complementary to that of Ushio et al (2018) and Grziwotz et al (2023) which we raise in the Discussion. We did not include those approaches as they are restricted by time series length (requiring >10 time points to function and >30 for reliable estimates) and fall under ‘resilience indicators’ rather than specifically early warning signals as our focus here. In our dataset, this time series length requirement would exclude Loch Leven, Monona and Washington from our analysis which we feel too greatly diminishes our sample size. We similarly believe Williamson and Lenton (2015)’s indicator has wider generic value than just as an EWS (due to its estimation of the Jacobian) but found it experienced a similar requirement for time series length (>15 time points in this case). We have therefore not added it to our analysis but have added references where appropriate to the method and a brief analysis in this response document for the reviewer’s interest. Lines 341-347.

‘For example, Ushio et al. ⁶³ exploit empirical dynamic modelling to estimate the Jacobian matrix of a multivariate community and extract a stability index which accurately diagnoses vulnerable periods in fish communities. This has been developed further by Medeiros et al. ⁵⁹ to identify key species for management based upon their contribution to the system’s Jacobian and Grziwotz et al. for univariate time series ⁶⁴. Similarly, Williamson and Lenton

⁶⁵ approach Jacobian estimation using multivariate autoregressive models with equivalent success.’

4) When the authors say that most published papers contain detected ecological transitions, this is an obvious effect of publication process: publishable are the results that contain a phenomenon of interest rather than those that do not. Publishing an absence of something is more difficult and is of marginal interest. I am sure the researchers in this community shelved lots of results where they detected nothing. I agree that such results can also be useful, but not necessarily for a broad research readership.

We would like to clarify that this argument is one not made by us (e.g. Barto, E. K. & Rillig, M. C. Dissemination biases in ecology: effect sizes matter more than quality. *Oikos* **121**, 228–235 (2012)) but is of particular relevance to early warning signal usage as the biasing of published results can increase risk of the ‘Prosecutor fallacy’ (Boettiger and Hastings, 2012, Early warning signals and the prosecutor's fallacy *Proc. R. Soc. B.* **279**, 4734–4739) if only systems known to undergo transitions are studied. It also represents a warning for practitioners who use regime shift and tipping point detection methods (i.e. EWSs) to classify thresholds of stress (Hillebrand, H. et al. Thresholds for ecological responses to global change do not emerge from empirical data. *Nat. Ecol. Evol.* **4**, 1502–1509 (2020)) for management purposes (a key target audience for this work). Thus, we use this as rationale for the inclusion of a range of lakes in this analysis showing a range of transition types.

5) I always start reading a new paper by looking at its figures to see the data under study, and then at the obtained results, to see if they are non-trivial. With this paper, I started searching for the main data and, to my surprise, found it in the supplement (figure S1). I think this figure should be in the main text. Then, I looked carefully at the top panels of figure S1 and placed the ticks in the panels where I could detect some transitions by eye. These were panels 1, 2, 6 and 8 (lakes Kasumiguara, Kinneret, Monona, and Washington). Curiously, after reading the paper in full, I found that the authors’ analyses detected critical transitions in the same datasets (Table S1). On the one hand, it is a good confirmation, on the other hand, it means that some less visible transitions may still be non-detectable by the applied methods. Therefore, modelled data is still of importance in studying such systems: it provides sufficient statistics and,

more importantly, ensures controlled experiments with ground-truth comparison. I think the authors should change the title of their paper, as in its current form it sounds dismissive to the above issues. The title could be “Analysis of EWS in empirical lake data”.

We agree that ideally Figure S1 would be presented in the main text, but we believed the size of figure would be inappropriate anywhere other than in the Supplementary. If the editor is willing for a whole page to be allocated to the figure, or the reviewer can suggest an elegant way of compressing the figure whilst retaining the visibility of the information, then we are happy to move it to the main text where we agree it should belong.

We apologise for our confusion, but we do not understand the concern of the reviewer regarding ‘**less visible transitions may still be non-detectable by the applied methods**’. To our knowledge, our approach to classifying regime shifts extends the techniques used by all previous attempts for empirical regime shift detection (e.g. *changeoint*, Gsell et al 2014, Bestelmeyer et al. 2011, Cianelli et al. 2004), and ensures all classification are made via model fit rather than human interpretation. Similarly, not all regime shifts were classified as critical transitions (e.g Kinneret zooplankton) despite there being a visible regime shift in the time series, implying that the subtle differences can be identified.

6) The paper from the start discusses ideas that are clearly defined only later (critical transitions and abrupt changes in lines 405-410, 415-420). I think such blocks of text should appear much earlier, in case if readers are not familiar with the topic.

The reviewer is in agreement with the reviewers #1 and #3 and so we have added an additional figure and a glossary (Box 1 - Figure 1) for this concern. We hope that this clarification will be sufficient.

7) When the authors distinguish “univariate, multivariate and machine learning” indicators, this may be misleading for people in the ML community, because they are aware of ML methods that can be both univariate and multivariate. It looks like the authors developed their own language in this context, but it is not widely accepted and is better to be avoided in a peer-review publication.

We realise this statement may be confusing for readers, and were referring here to the differences in early warning signal calculation (which share a theoretical background) rather than ML. We have therefore altered our language throughout to briefly introduce ML (line 108-110) and then refer to the specific univariate ML model EWSNet for the remainder of the manuscript.

‘Both univariate and multivariate machine learning models are possible, but to date, only univariate forms have been trained specifically for tipping point classification’^{35,36,38}

8) When the authors claim that they disentangle critical transitions from regime shifts (lines 292-295), this is well-known in the EWS community for years and usually is addressed by using different indicators/techniques. This is not a novel conclusion

We do not argue that the EWS community has not explored identifying critical transitions vs regime shifts in empirical time series – often using combinations of change point analyses (e.g Gsell et al. 2016) – but this paper expands that work by allowing non-linear but continuous relationships (via threshold GAMs) and by explicitly comparing against system behaviour in the state space. This therefore allows us to identify possible transitions mechanisms from solely descriptive time series. That is not to say experiments and modelling are not required to verify these classifications (Scheffer and Carpenter, 2003) but we can provide ‘best guesses’ using solely monitoring data, which is the holy grail of EWS research (Dakos et al 2012).

We have strengthened this point by now using all three of Scheffer and Carpenter (2003)’s ‘indicators of stable states’/Bestelmeyer et al. (2011)’s ‘analytical indicators’ in our classifications by introducing a probability density peak analysis in the new Figure 2 and updated Figure 4, and linked our hypothesised behaviour of their three indicators to each transition type defined in Figure 1.

9) It would be useful to include a table with full description of the variables, sampling rates, durations, etc. Such a summary is important for overview of data. Also, include information about number of points in the original and processed data (monthly, yearly).

The reviewer makes a good suggestion and we have added Table 1 which contains an overview for each lake of the sampling method, original sampling frequency, sampling depth, monitoring period, length of monthly time series post regime shift identification, length of yearly time series post regime shift identification, the number of available genera, and whether a described regime shift has previously been published.

'Table 1. Data and monitoring characteristics of lakes contributing to early warning signal assessments.

Lake	Sampling method	Original sampling frequency	Sampling depth (m)	Period	Monthly time series length contributing to assessments	Yearly time series length contributing to assessments	Number of genera	Regime shift identified in the literature
Lake Kasumigaura	Tube sampler and vertical net haul	Bi-weekly	0-5	Aug 1981-Dec 2018	342	30	33	✓⁵³
Lake Kinneret	Tube sampler, mix sampling and profile sampling	Weekly	0-40	Jan 1975-Dec 2015	289	25	31	✓⁵²
Loch Leven	Tube sampler and vertical net haul	Weekly	0-5	Feb 1992-Dec 2006	152	12	6	✗
Lake Mendota	Tube sampler and vertical net haul	Monthly	0-20	May 1995-Nov 2018	168	15	37	✓⁵⁴
Lake Monona	Tube sampler and vertical net haul	Monthly	0-20	Apr 1999-Dec 2017	130	12	35	✗
Lower Zurich	Tube sampler and vertical net haul	Monthly	0-135	Jan 1977-Dec 2009	332	28	40	✗
Upper Zurich	Tube sampler and vertical net haul	Monthly	0-36	Jan 1980-Nov 2000	209	17	62	✗
Lake Washington	Tube sampler and vertical net haul	Weekly	0-20	Jan 1962-Dec 1994	97	9	12	✓⁴⁸
Windermere	Tube sampler and vertical net haul	Bi-weekly	0-40	Jan 1979-Dec 2002	244	20	16	✗

10) I do not agree that non-recorded points should be replaced by zeros (lines 375-376).

This processing changes auto-correlations in the data, which is the basis of early warning signal indicators. Such pre-processing will change the indicator slopes.

We understand the reviewer's concern as it is likely we were not clear in the text, but the rationale for replacing non-recorded points with zeros is because those genera were below detection threshold during of recording. Consequently, that genus is not missing and should be considered 'not found' despite the search effort being constant across the sampling period.

This is equivalent to counting the abundances of a butterfly species along a transect but not observing any individuals despite the species being present in that habitat – generating a 0 count.

Similarly, if we do not replace unrecorded plankton densities with zeroes, missing values would prevent sufficient EWS assessments as large periods of NAs would be introduced into the time series. As EWSs assume equally spaced observations (Dakos et al 2012), many of the time series and dramatically decrease our coverage of each lake. Interpolation has been suggested to populate missing values in EWS research but due to the large periods of NAs and potential to alter variance and autocorrelations in the data, we believe interpolation is inappropriate.

This has been clarified on lines 376-379.

‘Unidentified and/or unnamed species were removed and if a species was not recorded on a sampling date, that species’ density was assumed to be zero. This assumption results from constant search effort being made on each sampling date and unrecorded species being below detection threshold on that specific date.’

11) It would be good to see a multi-panel figure of monthly data. In fact, detection of transitions in monthly data is more challenging and interesting, because noise would mask obvious transitions visible in yearly data – can the authors include such results? The provided panels with state variables are of less interest, in my opinion.

The reviewer raises a similar concern to reviewer #1 in comment 23) where we have answered in detail. In brief, we repeated the TGAM classification of regime shifts, critical transitions and stationary time series in the monthly data and found agreement in the breakpoint year (though, of course, in monthly time series, that breakpoint was between months rather than years).

12) In lines 113-115, the authors mention nine lake datasets and refer to figure 1, which, in fact, does not contain real data (there are schematics). The sentence should be modified.

We have now replaced Figure 1 with Figure 4 (due to the addition of other figures) which now contains real data. We hope this change satisfies the reviewer.

13) There are some terms and abbreviations that are not explained, like mgcv, brmc. I understand it is technical, but in that case, it can be in the supplement. There are acronyms that are defined multiple times (CSD, for example). It may be worth including a nomenclature.

We apologise for the lack of explanation for mgcv, brms and Stan specifically. These are the names of R packages/software and so have made this clearer by suffixing each time a new package name is introduced with '*package*'.

The multiple redefinition of CSD was to ensure that readers who may only read the main text and not the methods (or vice versa) would not have to trawl the remainder of the text. We have therefore taken the reviewer's advice, removed the redefinition and added a glossary (Box 1) to mitigate this.

14) In the Supplement, there are many plots with time series of chains (four overlapping records that are difficult to distinguish). I am not sure how useful is this, but it can remain in the supplement for information. The captions of figures S2-S7 mention monthly and yearly data, but it is not clear what lake is where – is data from all lakes combined? The y-axis labels are merged and unreadable. I think the label tick/label step should be changed when mapping the labels (there should be much fewer of them). In the right-hand-side panels, top labels are wider than the panels.

The figures S2-S7 are diagnostic plots for the Bayesian logistic models fitted to the yearly and monthly early warning signal data which validate the appropriateness and distribution of uncertainty for each estimated model parameter. They have been included for completeness. The chains in the right hand column represent the repeated sampling by the Monte Carlo Markov Chain (MCMC) sampler which we expect to overlap and 'mix' consistently throughout sampling. The left hand column depicts the estimated distribution for each model parameter which we expect to be smooth and unimodal. Consequently, figures S2-S7 show that all our models have converged and are generating robust parameter estimates (not necessarily accurate but precise and replicable from the data and prior). Figures S8-S10 allow

use to then confirm how well our model fits our data with high congruency between the observed data (dark lines) and draws from the model's posterior (light lines). The labels have now been reformatted to be readable.

As separate models were fitted for monthly and yearly EWS data, each supplementary figure is repeated; once for each model.

We would prefer to keep these supplementary figures as they are following good practice for Bayesian model reporting, but understand if the reviewer and editor believe they are unnecessary and will remove accordingly if requested.

15) In tables S3-S13, why are effective sample sizes are not integer? Why in these panels Rhat is included, which is equal to 1 everywhere, - isn't it better to mention this in the text or caption instead of dragging the same value in a separate column in all tables?

Effective sample sizes (ESS), as reported by the Stan language and the brms R package, represent the number of independent draws taken from the Monte Carlo Markov Chains (MCMC). These chains are stochastic processes that sample an assumed prior distribution, assign probabilities to that sample and then resample using the updated beliefs. ESS represents how well the sampler has estimated the parameter value given the number of samples/iterations the model has performed (Geyer, Charles J. 2011. "Introduction to Markov Chain Monte Carlo." In *Handbook of Markov Chain Monte Carlo*, edited by Steve Brooks, Andrew Gelman, Galin L. Jones, and Xiao-Li Meng, 3–48. Chapman; Hall/CRC.). The MCMC sampler can introduce error (and decrease the ESS) through autocorrelation within a chain and divergence between chains. The high ESSs reported in Tables S3-S13 can be interpreted as a validation that the model has converged and is an appropriate fit analogous to successful overdispersion, Cook's distance or F-tests for frequentist models.

Rhat meanwhile represents the ratio of the average sample variance within a chain versus the variance of the pooled samples across the separate chains (Gelman, Andrew, and Donald B. Rubin. 1992. "Inference from Iterative Simulation Using Multiple Sequences." *Statistical Science* 7 (4): 457–72.). If the chains have converged within the same parameter space (i.e. are at equilibrium) then the two variances are equal and Rhat equals 1. Rhat is therefore

another Bayesian measure of convergence and an appropriate model diagnostic criterion included for completeness.

16) The main text does not have page numbers, and the supplement has three pages numbered as 1.

Page numbers removed from the supplement accordingly.

Reviewer #3

Summary of work

The authors conduct an analysis of early warning signals in empirical lake data. They obtain plankton densities and abiotic drivers from 9 lakes that have publicly available data, and assess whether a critical transition occurs by fitting TGAMs and looking for break points in both the plankton and abiotic driver data. From a total of 244 time series, they find that 35 undergo a critical transition, by their definition. They test three different groups of early warning signals (EWS) on this data: univariate EWS; multivariate EWS; and EWSNet, a machine learning method. They also test a variety of different preprocessing steps (detrending and deseasoning), data resolutions (monthly and yearly averages), and EWS computation methods (rolling vs expanding window). They find that univariate EWS with an expanding window obtain the highest average performance in yearly data, and multivariate EWS with an expanding window obtain the highest average performance on monthly data. They found that other combinations of EWS were not much better than chance. They conclude that EWS based on bifurcation theory are not particularly useful in lake data, and that focus should turn to resilience indicators and methods that are specific to lake ecosystems.

Summary of evaluation

I applaud the others on conducting a thorough evaluation of a multitude of different EWS and preprocessing methods. It has already been demonstrated that EWS are not consistent in freshwater ecosystems (e.g. Gsell et al. PNAS, 2016), but this study builds

on this work by proposing a new way to identify critical transitions in lake data, increasing the number of lakes analysed, testing EWS on transitioning and non-transitioning time series, and testing a wider range of EWS, including a recently developed machine learning method (EWSnet). I've since taken a look at the data myself, and I don't think it's surprising that the EWS do not perform well, based on the number of missing data points, the seasonality, and the relatively small number of data points prior to the transitions, but nonetheless, I think its a useful contribution to the field, and an important word of caution to the use of generic EWS in empirical data.

I was able to download the code and data files from the Github repository, which is well documented. The code looks well organised, although I have not attempted to reproduce the results. The methods seem sufficiently detailed to follow the same steps as the authors. I think that the work is suitable for Nature Communications, although I have a few main comments and a few minor comments that I think should be addressed.

Main comments

1) The processed dataset includes 35 'transitioning' and 209 'non-transitioning' time series. It's important to bear in mind that this is an unbalanced dataset. If a classifier/EWS picked non-transitioning every time, it would have a very high correct prediction probability of $209/244=0.86$. Therefore I don't think that correct prediction probability (which is used in the results section line 190-204) is a good measure of performance. Something like the F1-score may be more appropriate, which strikes a balance between the true positive rate and the true negative rate. ROC curves are also a great way to get performance measures on a binary classification task (using the area under the curve). I would like to see which EWS perform best with a metric that is more suitable for an unbalanced dataset.

The reviewer highlights an important point which we aimed to circumnavigate using our multi-level modelling approach. By explicitly accounting for interdependencies in the EWS dataset (i.e. repeated measurements within a lake and indicators expected to behave similarly in transitioning vs non-transitioning systems), the probability of correct prediction is regularised and the confidence in the prediction shrinks and becomes wider. This is the key benefit our approach has over typical binary classification task metrics such as AUC or F1-

scores, as the test data set is ultimately imperfect, with confounders and interdependencies inescapable in natural, empirical systems. We also then tried to disentangle this further by splitting the overall analysis (Figure 3) in to true positive and true negative ability (Figure 4). Targeting EWS ability in this way can verify where each indicator performs best and worst analogously to AUC/F1-score with the overall analysis only intending to give an overview.

This being said, we repeated our analysis using both balanced accuracy and F1-scores. Both statistics reveal comparable estimates to our modelling approach (although with a single point estimate). However, following the reviewer’s concerns, we have added an additional weighting term for sample size which regularizes our probability estimates further, so that the initially extremely high scores reported in the original submission for expanding window methods in Yearly data are much more unconfident. This implies that F1-score is overweighting towards the transitioning data where expanding windows often signal but generate false positives in the non-transitioning data. We have consequently updated the results with the new model and provided balanced accuracy and F1-score results below. Panels A) and B) are direct analogues of Figures 5 and 6 from the main text.

2) It is mentioned throughout the manuscript that multivariate EWS outperformed univariate EWS (abstract, line 124, line 251, line 306). Can the authors explain how this statement is supported by the results? Line 190 seems contradictory to this: “univariate EWS estimated using expanding windows displayed the highest average probability of correct classification”. To me, what seems most noteworthy is the fact that the expanding window improves performance regardless of whether you use uni/multivariate data. This may be due to large number of missing data points in the plankton time series, and the relatively small amount of pre-transition data for a given lake.

The order of computation methods from top to bottom indicates the average ranking of each computation method across monthly and yearly data. This conclusion is therefore clearer in the updated analysis following the reviewers’ comments, with greater weighting towards the critically transitioning data. Now multivariate forms of each computation method have a higher average prediction probability than univariate though the differences are minimal. This has been clarified in the caption of Figures 5 and 6:

‘Computation methods are ranked by their mean ability across monthly and yearly data.’

3) The time series are split into ‘transitioning’ and ‘non-transitioning’. Where do the time series with a regime shifts but no critical transition go? I think into the ‘non-transitioning’ category, but this should be made more clear in the manuscript. If this is the case, when the authors compute EWS in these non-transitioning time series, are they including the section with the regime shift, or trimming the data to just before the regime shift? If the former, one would expect a spike in variance during the regime shift, which could trigger false positives in this analysis. I think it would make sense to also trim the regime shift (non-transitioning) time series so the regime shift is not included in the EWS computation.

The reviewer is correct that for our EWS ability analysis, only two classes were of interest with a relevance to regime shifts: critical transitions (expected to display CSD) and not critical transitions. Non-critical bifurcations are also expected to display CSD but none were identified in our lake dataset. This information has been added to the text on lines xx:

‘Many of the other lakes displayed breakpoints in their time series (e.g. Lake Mendota’s zooplankton) but these were not matched in the environmental state space (Figure 2, Figure 4A, Figure S1) and were therefore classified as abrupt non-bifurcations if they also displayed bimodality. There are therefore two primary classifications relevant to EWSs – critical transitions and not critical transitions. These classifications made by TGAMs were then used to ground truth downstream EWS assessments and trim time series to pre-transition data.’

In response to the second concern, time series with non- bifurcation regime shifts typically belonged to a lake where other time series displayed a critical transition. We therefore trimmed all time series in lake with critical transitions prior to the estimated time series breakpoint. Consequently, the regime shift should not be included in the EWS assessments as the reviewer recommends. This information is clarified in the text on line 514-519.

‘To enable comparability between transitioning and non-transitioning taxa, lakes containing transitions were subset prior to the year identified by TGAMs. Resultantly, if one of a lake’s trophic level experiences a critical transition whereas the other experiences a non-bifurcation abrupt shift, then all time series are subset prior to any regime shift. This minimises the likelihood of false positive signals driven by the changes in variance

experienced in non-bifurcation regime shifts. Lakes with no regime shifts were subset to 85% of their total length. This ensures we can infer the near future of the non-transitioning lake correctly.'

4) The Github repository is well organised and well documented. However, I didn't find any indication as to what each data file represents? There are 4 files in the data directory. I suggest that the authors indicate this somewhere in a readme file.

We apologise for the lack of description for the repository data files. Two of the files were duplications of the main plankton density data and have been removed. A README has now been added which reads:

*'transition_dates.csv - estimated critical transition dates generated by the threshold generalised additive models coded in lake_state_spaces.R.
wrangled_genus_plank_data.Rdata - the cleaned and genus aggregated plankton data underpinning the early warning signal assessments.
The remainder of the early warning signal data can be found in the Results folder.'*

Changes have also been made to the published files and directory following additional analyses requested during this resubmission.

5) It's not clear to me what the difference is between the scaled and unscaled weights of the ML classifier. Please explain this somewhere, and why it has such a large impact on the EWSNet predictions.

Both scaled and unscaled weights are derived from the same training dataset but with different pre-processing. The "unscaled" involved training on the raw training data which spanned three orders of magnitude (1,10,100) whereas "scaled" normalises each time series within the range [1-2] using the equation:

$$s = 1 + \frac{x - x_{min}}{x_{max} - x_{min}}$$

where x is the time series to be scaled.

This information has been included in the Methods section on lines 504-512.

‘We additionally tested the effect of scaled versus unscaled data processing on the quality of EWSNet predictions. The scaled model involves training on the same data as the unscaled, but time series were normalised between within the range [1-2] using the following equation:

$$s = 1 + \frac{x - x_{min}}{x_{max} - x_{min}}$$

where x is the training timeseries. This scaling therefore ensures all dynamics are considered at the same magnitude and aims to minimise the impact of measurement scale on predictions. When testing using the scaled form of EWSNet, the test time series must also be scaled for appropriate predictions.’

6) Am I right in thinking that break points were computed using total plankton densities? Did the authors consider EWS in total plankton densities instead of individual densities? Given that there were many zeroes in the individual plankton time series, I'd be curious to know if EWS have higher performance on the aggregated time series.

The reviewer is correct that breakpoints were estimated using trophic level data while EWS assessments were made at the genus level. We have responded to this concern in response to reviewer #1's comment 5).

We had not performed EWS assessments for total densities as multivariate EWSs require multiple time series which is not possible for total densities. However, we have attached here a brief analysis for the univariate EWS and EWSNet indicators for the reviewers' interest.

The overall results are consistent other than scaled EWSNet displaying more robust true positive behaviour (relative to genus level data). Expanding windows display worse true negative capability in monthly data presumably due to stronger autocorrelation in overall system dynamics. Please note that analysis is of a very small sample size (4 critically transitioning vs 22 not critically transitioning).

7) There is more recent relevant work on machine learning for EWS that could be cited on line 315:

Patel and Ott, Using machine learning to anticipate tipping points and extrapolate to post-tipping dynamics of non-stationary dynamical systems, *Chaos* 2023.

Dylewsky et al., Universal early warning signals of phase transitions in climate systems, *Interface*, 2023.

Cited where appropriate. Line 110.

8) Fig 1B: what data is being shown here? Real lake data?

This data is not real data but an exemplary multi-species community undergoing a critical transition/fold bifurcation provided by the R package *EWSmethods*. As Figure 1 is an overview schematic, we did not originally use real world data, but following the suggestions of Reviewer's #1 and #2, we have replaced all simulated/schematic data with real lake data. This is now presented in Figure 4.

9) How is the scaled metric score in Figure 2 computed? I didn't see it in the methods -

sorry if I missed it.

Scaled metric score is simply the plankton density normalised via the equation:

$$s = \frac{x - \bar{x}}{\sigma_x}$$

where x is the plankton time series, \bar{x} is the time series mean and σ_x is the time series standard deviation. The only purpose of this scaling is to improve the ease of plotting and comparison between trophic levels and models. We have added this information to the Figure legend (Figure 4 and Figure S1).

'... A) Application of the classification approaches introduced in Figure 2 in the yearly lake plankton data. An example of an abrupt, non-bifurcation shift (Lake Washington's zooplankton), a critical transition (Lake Kinneret's phytoplankton) and a non-transition (Windermere's phytoplankton) are presented. Plankton densities have been scaled to mean zero and unit variance to improve plotting clarity....'

10) Line 62: CSD does not increase as such, it is the phenomena of an increasing return time following perturbations.

The sentence has been edited to better convey this point. Line 64-66.

'EWSs attempt to detect this critical point by the phenomenon of Critical Slowing Down (CSD) or the increasing return time to equilibrium following perturbations as a critical transition is approached ¹⁰'

11) Abstract: “recently developed machine learning techniques”. I think this is too broad, since only a single machine learning technique was tested. I think it should read “a recently developed machine learning technique”.

Changed accordingly.

12) Line 60: “bifurcation theory which states that a....” → “bifurcation theory which describes how a...”

Changed accordingly.

13) Line 102: “machine learning exploits”

Changed accordingly.

14) Line 753: “machine learning is limited to univariate time series” → “machine learning is applied to univariate time series” (there are techniques to apply it to multivariate time series)

Changed accordingly.

15) Figure 2: x-axis “Explanatory variable”

Changed accordingly.

REVIEWERS' COMMENTS

Reviewer #1 (Remarks to the Author):

In the revised version of the manuscript: "Early warning signals require critical transitions in empirical lake data", the authors have diligently worked on including the reviewer's comments. In my opinion, this has improved the manuscript considerably. The more stringently used terminology and much-improved method part make the manuscript much easier to read, follow and reproduce.

However, two points of criticism remain. These regard ambiguities in Figures 1 and 2 and, in this connection, the authors assumption of the capabilities of TGAMs.

First, I very much appreciate the general idea of Figures 1 and 2 to clarify the theoretical background and hypotheses and define the used terminology for the paper's scope. But I think both figures could improve. In Figure one should be more evident when the authors use time-space or phase-space characteristics to derive their grouping. I also could not follow the logic of why step changes in the state variable in time-space due to a large shift in control parameter qualify as rapid regime shift, while a non-linear trend (i.e. State threshold following the terminology of Andersen et al.) is classified under gradual, smooth transitions, particular since in a matching step in time-space in driver and state variable results in a linear trend in phase-space (also mentioned in D). Also, under the label abrupt non-bifurcation, the example of cyclic regime shifts is given; however, in this generality, cyclic regime shifts might well involve bifurcation points and thus possibly EWSs, see, for instance, Scheffer and Carpenter 2003, Darkos et al. 2014.

Further, I find the figure caption could be improved. For instance, I found it confusing that the authors open with, "In its simplest form, a regime shift is the process whereby an ecosystem rapidly changes from one alternative stable state to another ..." which is commonly the wording used to describe critical transitions, they then continue "whereas critical transitions .." where they again refer to critical transitions. I would also doubt that the existence of a positive feedback mechanism is easy to demonstrate based on data; at least, it is nothing that the authors did. On the other hand, characteristics like hysteresis and discontinuity are missing, which are later used by the authors in their line of argumentation. Further in the glossary, the authors define regime shifts as sudden or abrupt shifts to an alternative attractor; however, this technically does not include the case of matching step changes in driver and state variables (B), so overall, this definition appears inconsistent. I would have also appreciated it if the authors had found a minimal set of needed terminology for the scope of their manuscript and used this consequently; for example, they define bifurcation and use the word tipping point within the definition. Thereof follows an additional definition for tipping point without clearly stating the subtle difference to bifurcation. As I understood it, tipping point was used as a more specific term for the particular case of critical transitions; however, when the authors define critical transitions, they use the term critical value/ bifurcation point instead. Also, it would have been helpful to introduce the words discontinuity and hysteresis here. Further, the given definition of a smooth transition clearly subsumes the case of matching breakpoints in time space and an overall linear relation in phase space, which is also referred to in class B.

In regard to Figure 2, I find the most important and maybe most difficult case to distinguish is not clearly depicted, i.e. the non-linear smooth transition (state threshold following Andersen et al.). From a depiction of this case, it would have become clear that one would also expect a bimodal distribution there. Thus, the only evidence remaining to distinguish state thresholds from critical transitions is hysteresis evidenced by overlapping data clouds and more than one breakpoint in phase-space. In cases where there is no overlap, the remaining difference might be the presence of discontinuity, which is however very hard to detect under common monitoring data quality.

This leads me to the potential of TGAM, where I sincerely doubt that they can distinguish between State threshold cases and driver-state hysteresis in the absence of an indicator for hysteresis, i.e. overlap of data clouds and more than one threshold, and to my knowledge, there exists no demonstration in regard to this capabilities. In L 400- L 405, the authors mention overlap as a criterion, however, hysteresis can not be discerned from bimodality and seems otherwise not to have been used in the classification of critical transitions and non-critical transitions.

The authors have already toned down the certainty in their classification, e.g. figure 1 caption and added a statement in the discussion (L278 - L281), however, I would have sincerely wished that the authors were much clearer and more consistent about these limitations throughout the text, e.g. that they would have refrained from statements like “the precise classification” (L125) or be clearer in figure 2 (see comments above). I do not think these well-known problems have a major impact on the assessment of the true positive abilities of EWSs since the authors made a conservative selection there. On the other hand, this conservative choice might have introduced cases with EWS into the class of non-critical transitions and thus might have affected the true negative predictions, depending on the number of ambiguous cases, unfortunately, the authors do not quantify this. In the light that these are well-known technical and thus acceptable problems, I would have much appreciated a very clear communication or an exclusion of these ambiguous cases altogether.

Some remaining issues are:

- In Figure 2, fourth row, second column subfigure (4,2): From the subfigure in the fourth row, the first column (4,1), it occurs that there are no flat-tail high system state values, so the curve in 4, 2 should have either an L shape or L shape mirrored at the y axis.

- L197: “This results ...” should be “These results....” or “The results ...”

- L276: This might refer to Figure 4, not 2.

Reviewer #2 (Remarks to the Author):

The authors produced a substantial revision, and the revised manuscript reads better.

I still have two comments.

1) I disagree with the new title "Early warning signals require critical transition..."

This is logically incorrect, as an attribute cannot "require" anything from its process. In my review, I suggested title "Analysis of EWS in empirical lake data". If the authors want to stress that EWS is an attribute of critical transitions (which is actually not a new message), they may say "Critical transitions in empirical lake data are accompanied by EWS" (or "have EWS signatures")

2) I still think that in tables S3-S13 (eleven tables, each of one page) the columns of R_{hat} with repeated values 1, all the same and not informative in such quantities, must be removed, and just a line with this information ($R_{hat}=1$) should be added to each table caption.

Reviewer #3 (Remarks to the Author):

The authors have addressed my comments and concerns and significantly improved the figures in the manuscript.

Reviewer responses

Reviewer #1

In the revised version of the manuscript: "Early warning signals require critical transitions in empirical lake data", the authors have diligently worked on including the reviewer's comments. In my opinion, this has improved the manuscript considerably. The more stringently used terminology and much-improved method part make the manuscript much easier to read, follow and reproduce.

However, two points of criticism remain. These regard ambiguities in Figures 1 and 2 and, in this connection, the authors assumption of the capabilities of TGAMs.

We thank the reviewer for their direction in improving the clarity and technical details of the manuscript. We believe we have addressed their remaining comments below.

In the revised version of the manuscript: "Early warning signals require critical transitions in empirical lake data", the authors have diligently worked on including the reviewer's comments. In my opinion, this has improved the manuscript considerably. The more stringently used terminology and much-improved method part make the manuscript much easier to read, follow and reproduce.

However, two points of criticism remain. These regard ambiguities in Figures 1 and 2 and, in this connection, the authors assumption of the capabilities of TGAMs.

1) First, I very much appreciate the general idea of Figures 1 and 2 to clarify the theoretical background and hypotheses and define the used terminology for the paper's scope. But I think both figures could improve. In Figure one should be more evident when the authors use time-space or phase-space characteristics to derive their grouping.

We thank the reviewer for their encouragement of Figures 1 and 2 and we have edited them to include their suggestions. We do not believe we are in disagreement with the reviewer for the majority of their points, and it is the poor wording on our part that has limited the current figures. We hope the reviewer finds them improved.

We have now split Figure 1 in half and labelled whether time-space characteristics or phase-space characteristics are used to distinguish groupings. In brief, the distinction between ‘transition-no transition’ and ‘abrupt-gradual’ involves the time-space whereas the subcategories involve the phase-space.

2) I also could not follow the logic of why step changes in the state variable in time-space due to a large shift in control parameter qualify as rapid regime shift, while a non-linear trend (i.e. State threshold following the terminology of Andersen et al.) is classified under gradual, smooth transitions, particular since in a matching step in time-space in driver and state variable results in a linear trend in phase-space (also mentioned in D).

We agree that the ‘threshold-like’ dynamics described in Andersen et al. 2009 is a member of the abrupt transition and non-bifurcation class (class B) rather than the smooth transition class. We have therefore added it as an example in B as a replacement for cyclic transitions (see response to comment 3).

We also apologise for the vague terminology of examples in the smooth transition (class D), which we did not intend to include threshold-like dynamics. Instead, we were referring to

Kefi et al.'s (2013) gradual curve that qualitatively describes exponential decay. We have therefore clarified this example in Figure 1 and added it to Figure 2 - Smooth transition (b - non-linear).

3) Also, under the label abrupt non-bifurcation, the example of cyclic regime shifts is given; however, in this generality, cyclic regime shifts might well involve bifurcation points and thus possibly EWSs, see, for instance, Scheffer and Carpenter 2003, Dakos et al. 2014.

The reviewer highlights a valid point regarding cyclic regime shifts as the mechanism may differ between different examples. Dakos et al. (2015) highlight an example where the driver

continually increases and relaxes to allow both critical transitions and non-bifurcations to cause successive regime shifts, whereas Rinaldi and Scheffer (2000) describe how the bistable region of the Hopf bifurcation may also manifest in cyclic shifts. Cyclic shifts as a blanket phenomenon consequently do not fall in to any one of our Figure 1 classifications though each individual shift should independently be classifiable using Figures 1 and 2.

We have therefore removed cyclic regime shifts from abrupt non-bifurcations as an inappropriate example and replaced it with the threshold-like responses the reviewer encouraged in comment 2).

4) Further, I find the figure caption could be improved. For instance, I found it confusing that the authors open with, "In its simplest form, a regime shift is the process whereby an ecosystem rapidly changes from one alternative stable state to another ..." which is commonly the wording used to describe critical transitions, they then continue "whereas critical transitions .." where they again refer to critical transitions. I would also doubt that the existence of a positive feedback mechanism is easy to demonstrate based on data; at least, it is nothing that the authors did.

We have altered the terminology in the revision to avoid stable states and to focus on fundamental changes (following Dakos et al. 2015 and Andersen et al. 2009). This then fulfils the use case of many practitioners. It then distinguishes regime shifts from critical transitions and other mechanisms that explicitly require the theory of alternative stable states. For example, Dakos et al. 2015 highlight how it is the abrupt and persistent nature of regime shifts that is typically of interest to many researchers and managers. They then describe the various mechanisms that can generate such regime shifts which we attempt to characterise here.

'Regime shift: Sudden or abrupt shift in the state of the system resulting from the influence of an external control parameter/driver or by the system's internal dynamics, where core ecosystem functions, structures and processes are fundamentally changed. A regime shift may be associated with bifurcations (after crossing control parameter thresholds/tipping points), step changes in state (in response to step changes in control parameter), threshold-like responses (sigmoidal response to control parameter), or limit cycles (cyclic changes due to

the system's internal dynamics). These abrupt shifts may also occur across different trophic levels.'

Similarly, we agree that existence of a positive feedback mechanism is extremely challenging (and possibly impossible) from observational data alone. It is however a key feature of critical transitions and so we are required to mention it here, even if current techniques can't identify it. We have caveated this in the Figure 2 caption.

'TGAMs are limited by classifying system dynamics solely upon observational data and therefore will not guarantee classification without knowledge of the underlying system equations. Those equations can only be determined through experiments and differential equation modelling ²⁴, but TGAMs provide a 'best-guess' using the limited data typically available to system managers.'

5) On the other hand, characteristics like hysteresis and discontinuity are missing, which are later used by the authors in their line of argumentation.

Hysteresis and discontinuity are now important characteristics for defining critical transitions in the Glossary (Table 1).

6) Further in the glossary, the authors define regime shifts as sudden or abrupt shifts to an alternative attractor; however, this technically does not include the case of matching step changes in driver and state variables (B), so overall, this definition appears inconsistent. I would have also appreciated it if the authors had found a minimal set of needed terminology for the scope of their manuscript and used this consequently; for example, they define bifurcation and use the word tipping point within the definition. Thereof follows an additional definition for tipping point without clearly stating the subtle difference to bifurcation. As I understood it, tipping point was used as a more specific term for the particular case of critical transitions; however, when the authors define critical transitions, they use the term critical value/ bifurcation point instead.

We have replaced the attractor terminology with 'fundamental changes' to be inclusive of step changes in driver and state variables.

Similarly, bifurcation point is now defined and referenced alongside tipping point to highlight the subtle difference highlighted by the reviewer.

‘Tipping point: A threshold value at which a dynamical system undergoes a sudden shift from one stable state to another alternative stable state in response to small stochastic perturbations (Scheffer et al. 2012).

Bifurcation point: A threshold value specifically associated with a bifurcation (Scheffer et al. 2012; Boettiger et al. 2013).’

7) Also, it would have been helpful to introduce the words discontinuity and hysteresis here.

These terms have now been added or mentioned in the glossary (Table 1).

8) Further, the given definition of a smooth transition clearly subsumes the case of matching breakpoints in time space and an overall linear relation in phase space, which is also referred to in class B.

We agree and have responded to this comment in response to comment 2).

9) In regard to Figure 2, I find the most important and maybe most difficult case to distinguish is not clearly depicted, i.e. the non-linear smooth transition (state threshold following Andersen et al.). From a depiction of this case, it would have become clear that one would also expect a bimodal distribution there. Thus, the only evidence remaining to distinguish state thresholds from critical transitions is hysteresis evidenced by overlapping data clouds and more than one breakpoint in phase-space. In cases where there is no overlap, the remaining difference might be the presence of discontinuity, which is however very hard to detect under common monitoring data quality.

We entirely agree with the reviewer that the threshold-like vs critical transition dynamics are the most challenging to disambiguate, and that hysteresis is the key feature to distinguish them. We have therefore added to Figure 2 sub-classification examples for abrupt transitions to describe (similar to Andersen et al. 2009) how threshold-like transitions are anticipated to

behave in the threshold GAM framework – i.e non-linear changes in both the time and phase spaces, a bimodal distribution and no hysteresis.

10) This leads me to the potential of TGAM, where I sincerely doubt that they can distinguish between State threshold cases and driver-state hysteresis in the absence of an indicator for hysteresis, i.e. overlap of data clouds and more than one threshold, and to my knowledge, there exists no demonstration in regard to this capabilities. In L 400-L 405, the authors mention overlap as a criterion, however, hysteresis can not be decerned from bimodality and seems otherwise not to have been used in the classification of critical transitions and non-critical transitions.

We do not endorse TGAMs as a panacea for regime shift detection due to the challenges the reviewer rightly highlights, but we do believe that their flexibility allows us to describe more forms of dynamics than the typically applied linear techniques to regime shift detection (STARS, segmented regression, bifurcation baby form modelling etc). Consequently, we believe TGAMs are the current ‘best guess’ available for generic regime shift classification from observational data, but fully support the importance of experimental and modelling systems for unambiguously classifying systems as strongly endorsed by Scheffer and Carpenter 2003. This is highlighted on lines 168-179:

‘The time series classifications performed by TGAMs represent insights into the likely mechanism of change, covering a range of regime shift relevant mechanisms. There are, however, certain mechanisms that cannot be disambiguated without experimental or simulated work due to their similar behaviour across time and state-spaces. For example, threshold-like responses⁵⁴ and cusp bifurcations⁷ will both display sigmoidal responses in both time and state-space (Figure 2 – b) threshold-like) but only the cusp bifurcation is anticipated to exhibit CSD. For this study, critical transitions are sufficiently different from other mechanisms to be classified (assuming some relaxation of driver has occurred to evidence hysteresis) compared to non-bifurcation regime shifts, but we suggest that for qualitatively similar mechanisms, further evidence is necessary. Experimental and modelling to identify plausible system equations is an appropriate avenue to supplement TGAMs fit to observational data only.’

And lines 424-430:

‘Together, these three analyses allow us to disentangle critical transitions from other forms of non-bifurcation regime shifts such as pulse events or step changes, while also identifying non-linear but continuous transitions (i.e. non-critical transitions and smooth transitions), not feasible without the use of GAMs. That said, TGAMs are descriptive of observational data rather than diagnostic, and true classification requires some understanding of the governing system equations not achievable from observational data alone.’

In this revision, we have been more explicit in the description column of Figure 2 for the requirements for distinguishing between dynamics and have caveated all our classifications further in the figure legend.

Figure 2 Caption

‘TGAMs are limited by classifying system dynamics solely upon observational data and therefore will not guarantee classification without knowledge of the underlying system equations. Those equations can only be determined through experiments and differential equation modelling ²⁴, but TGAMs provide a ‘best-guess’ using the limited data typically available to system managers.’

That said, regarding the distinguishing of critical transitions from threshold-like dynamics, we do anticipate that, if the sampling resolution was sufficiently high (relative to the scale that the transition is occurring), that continuity would be identifiable in the threshold-like response regime shift using TGAMs (see Figure 2), while discontinuity would be identifiable in critical transitions. We also agree hysteresis is key but that TGAMs can identify hysteresis from the overlap in smooths before and after the breakpoint. In fact, all four of the trophic levels we classify as critical transitions (Kasumiguara zooplankton, Kinneret phytoplankton and Monona zooplankton, Washington phytoplankton) all display overlap (Figure S1) which we have interpreted as hysteresis. We have interpreted these as hysteresis because, while the system has not shifted back to its original state, it has re-experienced abiotic driver values present prior to regime shift. In the threshold-like dynamics scenario, the system would have shifted back to the original state.

We have clarified this interpretation on lines 301-315:

‘...This is compounded further as the disambiguation of critical transitions from certain other regime shift mechanisms can be extremely complicated in empirical data^{24,25,54}, and the classifications we have made here are ultimately a ‘best guess’ given the data availability. For example, critical transitions (Figure 1 - A) and threshold-like responses⁵⁴ (Figure 1 – B) likely display identical time series, bimodality, and very similar state-space behaviour. The primary difference between the two mechanisms identifiable from empirical data is the presence of hysteresis which can only be observed if the system reverts entirely (i.e. regime shifts back to the original state) or partially (i.e the driver relaxes back in to the bistable region but not sufficiently for the system to shift back). As described in Figure 2, hysteresis can be identified by an overlap of TGAM smooths in the state-space, while a threshold-like response has no overlap. All the lakes we classify as critical transitions do display some degree of overlap/hysteresis (Figure S1), but other lakes’ breakpoints may not have sufficiently reverted for hysteresis to be identified. We therefore encourage empirical regime shift and EWS researchers to consider the mechanisms driving shifts^{1,24,25} to maximise their reliability and appropriateness, and to not solely use EWSs as evidence of approaching tipping points.’

11) The authors have already toned down the certainty in their classification, e.g. figure 1 caption and added a statement in the discussion (L278 - L281), however, I would have sincerely wished that the authors were much clearer and more consistent about these limitations throughout the text, e.g. that they would have refrained from statements like “the precise classification” (L125) or be clearer in figure 2 (see comments above). I do not think these well-known problems have a major impact on the assessment of the true positive abilities of EWSs since the authors made a conservative selection there. On the other hand, this conservative choice might have introduced cases with EWS into the class of non-critical transitions and thus might have affected the true negative predictions, depending on the number of ambiguous cases, unfortunately, the authors do not quantify this. In the light that these are well-known technical and thus acceptable problems, I would have much appreciated a very clear communication or an exclusion of these ambiguous cases altogether.

We have removed the reference to ‘precise’ throughout the text (replacing ‘precise’ with ‘explicit’ where necessary as we believe it is important to highlight that some consideration of possible dynamics is required when testing EWS ability), and have introduced an earlier

section discussing the ambiguous cases surrounding identifying hysteresis, threshold-like transitions and critical transitions in response to Comment 10) on lines 162-179.

‘Many of the other lakes displayed breakpoints in their time series (e.g. Lake Mendota’s zooplankton) but these were not matched in the environmental state space nor fulfilled the other requirements (e.g. no overlap of clusters to indicate hysteresis, Figure 2, Figure 4A, Figure S1) and were therefore classified as abrupt non-bifurcations if they also displayed bimodality. There are therefore two primary classifications relevant to EWSs – critical transitions and not critical transitions. Classifications made by TGAMs were then used to ground truth downstream EWS assessments and trim time series to pre-transition data. The time series classifications performed by TGAMs represent insights into the likely mechanism of change, covering a range of regime shift relevant mechanisms. There are, however, certain mechanisms that cannot be disambiguated without experimental or simulated work due to their similar behaviour across time and state-spaces. For example, threshold-like responses⁵⁴ and cusp bifurcations⁷ will both display sigmoidal responses in both time and state-space (Figure 2 – b) threshold-like) but only the cusp bifurcation is anticipated to exhibit CSD. For this study, critical transitions are sufficiently different from other mechanisms to be classified (assuming some relaxation of driver has occurred to evidence hysteresis) compared to non-bifurcation regime shifts, but we suggest that for qualitatively similar mechanisms, further evidence is necessary. Experimental and modelling to identify plausible system equations is an appropriate avenue to supplement TGAM fits to observational data only.’

We agree that our approach is conservative, but we are unsure how to approach the suggestion of communicating true and false positive rates as we truly do not know the mechanism that defines the dynamics of these lakes. We consequently cannot accurately fulfil this request and are using TGAMs as a ‘best guess’ for the dynamics prior to using EWSs; a classification typically lacking/unjustified from the previously published literature of natural systems.

12) In Figure 2, fourth row, second column subfigure (4,2): From the subfigure in the fourth row, the first column (4,1), it occurs that there are no flat-tail high system state values, so the curve in 4, 2 should have either an L shape or L shape mirrored at the y axis.

We have changed this accordingly, matching the direction of state vs both time and abiotic driver, but do want to highlight that we have not specified whether driver is increasing or decreasing through time, and so the state space plot could be in either orientation. I.e. if driver is decreasing through time and causes a transcritical transition, then the original schematic would be correct. However, for clarity, we have altered our figure and the caption to indicate that we are assuming abiotic driver is increasing through time.

‘Hypothesised behaviour of possible system dynamics under three complementary analyses used to classify the fate of a time series. These analyses fulfil the criteria of Scheffer and Carpenter²⁴, Andersen et al.⁵⁴, and Bestelmeyer et al.²⁵ for identifying alternative stable states in empirical data through i) time series shifts , ii) a hysteresis response to the control parameter and iii) multimodal distributions. We have assumed here that the control parameter/environmental driver is increasing through time. Analyses i) and ii) are performed using threshold generalised additive models (TGAMs) of plankton density against time and environmental driver respectively...’

13) L197: “This results ...” should be “These results....” or “The results ...”

We apologise for the confusion as this sentence was referring to why multivariate EWSs estimated using expanding windows had the highest average classification probabilities. We have clarified this with the sentence now reading:

‘This probability is associated with these EWSs displaying the highest probabilities in monthly data...’

14) L276: This might refer to Figure 4, not 2.

Changed accordingly.

Reviewer #2

15) I disagree with the new title "Early warning signals require critical transition..." This is logically incorrect, as an attribute cannot "require" anything from its process. In my review, I suggested title "Analysis of EWS in empirical lake data". If

the authors want to stress that EWS is an attribute of critical transitions (which is actually not a new message), they may say "Critical transitions in empirical lake data are accompanied by EWS" (or "have EWS signatures")

We thank the reviewer for their suggestion. We have followed their suggestion in combination with the editor to rename the title as: *Early warning signals have limited applicability to empirical lake data.*

16) I still think that in tables S3-S13 (eleven tables, each of one page) the columns of R_{hat} with repeated values 1, all the same and not informative in such quantities, must be removed, and just a line with this information ($R_{hat}=1$) should be added to each table caption.

We have fulfilled the reviewer's request by removing the R_{hat} column and adding that information to the tables' captions.

' R_{hat} was equal to 1 for all estimates.'

Reviewer #3

17) The authors have addressed my comments and concerns and significantly improved the figures in the manuscript.

We thank the reviewer for their useful comments and believe the manuscript is stronger and more useful as a result.